# Dopamine prediction error signaling in a unique nigrostriatal circuit is critical for associative fear learning

Daphne Zafiri[1,2], Ximena I. Salinas-Hernández[1,2], Eloah S. De Biasi[1], Leonor Rebelo[1] & Sevil Duvarci [1] ✉

Learning by experience that certain cues in the environment predict danger is crucial for survival. How dopamine (DA) circuits drive this form of associative learning is not fully understood. Here, in male mice, we demonstrate that DA neurons projecting to a unique subregion of the dorsal striatum, the posterior tail of the striatum (TS), encode a prediction error (PE) signal during associative fear learning. These DA neurons are necessary specifically during acquisition of fear learning, but not once the fear memory is formed, and are not required for forming cue-reward associations. Notably, temporally-precise inhibition or excitation of DA terminals in TS impairs or enhances fear learning, respectively. Furthermore, neuronal activity in TS is crucial for the acquisition of associative fear learning and learning-induced activity patterns in TS critically depend on DA input. Together, our results reveal that DA PE signaling in a non-canonical nigrostriatal circuit is important for driving associative fear learning.

Associative fear learning ('threat learning'[1]) — the ability to associate stimuli with threats — enables animals to predict and avoid danger and hence is critical for survival. However, learned fear that is excessive can also be maladaptive and have pathophysiological consequences. Much evidence indicates that anxiety disorders, such as post-traumatic stress disorder (PTSD), result from dysregulation of normal fear learning mechanisms[2–4]. Therefore, elucidating the neural mechanisms underlying fear learning is critical for understanding the pathophysiology of anxiety disorders and thus has high clinical significance. In the laboratory, this kind of associative learning is commonly studied using Pavlovian fear conditioning (FC), in which an initially neutral stimulus (conditioned stimulus, CS), typically a tone is paired in time with an aversive unconditioned stimulus (US), such as a mild electrical foot shock. As the CS-US association is learned, the CS acquires the ability to elicit fear responses that are associated with the US (such as behavioral freezing) so that it can elicit conditioned fear responses when later presented alone. Traditionally, the amygdala, particularly its lateral nucleus (LA), has been recognized as the primary brain region for acquiring CS-US associations during FC[5–10]. However, recent studies suggest that plasticity in brain structures beyond the canonical amygdala circuitry is also involved in acquisition of fear memories. Of note, the posterior tail of the dorsal striatum (TS) has recently been indicated in fear conditioning[11,12].

TS is a unique subregion within the dorsal striatum that receives a combination of inputs from structures such as the auditory, visual and rhinal cortices as well as the amygdala, that sets it apart from other dorsal striatal subregions[13,14]. Importantly, TS also receives dopaminergic innervation from a unique subpopulation of midbrain dopamine (DA) neurons that are predominantly located in the substantia nigra lateralis[15] (SNL). These DA neurons exhibit a distinct input-output organization compared to other midbrain DA neurons, project selectively to TS, and do not overlap with the subpopulations that project to the rest of the striatum, cortex and the amygdala[15]. Notably, recent studies have shown that these DA neurons are particularly important for novelty-induced threat avoidance[16,17], and they have also been shown to be involved in fear learning[12]. However, the precise role TS-projecting DA neurons play during associative fear learning is incompletely understood[18].

[1]Institute of Neurophysiology, Neuroscience Center, Goethe University, Frankfurt, Germany. [2]These authors contributed equally: Daphne Zafiri, Ximena I. Salinas-Hernández. ✉e-mail: duvarci@med.uni-frankfurt.de

Associative learning is driven by prediction errors (PE) that signal the discrepancy between predicted and actual outcomes[19]; and new learning happens when outcomes do not match expectations. It is well-established that ventral midbrain DA neurons, located in the ventral tegmental area (VTA) and the substantia nigra (SN), encode reward prediction errors (RPE) which act as teaching signals to drive reinforcement learning[20–24]. Recent studies have further demonstrated that ventral midbrain DA neurons encode positive PE signals not only for rewards, but also for omission of aversive outcomes[25–28] to drive associative learning. Interestingly, dorsal tegmental DA neurons that project to the amygdala have recently been shown to encode a PE signal to mediate fear learning[29]. However, although ventral midbrain DA neurons have been shown to play critical roles in aversion and fear learning[30–33], whether they contribute to PE signaling necessary for driving fear learning has remained elusive. Importantly, whether DA neurons projecting to brain structures outside the canonical amygdala circuitry encode a PE signal that is required for driving associative fear learning is largely unknown.

In this study, we investigate the precise role DA projections to TS play during associative fear learning. By performing measurements of DA terminal activity as well as DA release in TS, we found that DA projections to TS encode a PE signal for the aversive outcome during associative fear learning. Selective lesioning of TS-projecting DA neurons demonstrated that these neurons are important specifically during acquisition of fear learning, but not once the CS-US association was learned. Notably, the activity of these DA neurons was required for associating sensory cues with the aversive US, but not the context with aversive US or the sensory cues with reward. Conversely, temporally-precise optogenetic manipulation of DA terminals in TS during FC demonstrated that this PE signal drives fear learning and boosting this signal enhances associative fear learning. To gain further insights into the functional role of TS, we performed Ca²⁺ recordings of TS activity and found a PE-like activity pattern, as well as potentiation of CS responses, during associative fear learning. Bidirectional manipulations of TS activity further showed that the neuronal activity in TS is required for fear learning. Finally, we demonstrated that DA input was necessary for the fear learning-induced activity patterns in TS during FC. Taken together, our results reveal a key role for DA PE signaling in a unique nigrostriatal circuit for driving associative fear learning.

## Results

### DA neurons projecting to TS encode a PE signal during associative fear learning

In order to investigate the activity of DA neurons projecting to TS in a projection-specific manner, we used fiber photometry to measure activity-dependent Ca²⁺ signals at the terminals of DA neurons in TS. We injected a Cre-dependent adeno-associated virus (AAV) expressing the genetically encoded Ca²⁺ indicator GCaMP in SN of transgenic mice expressing Cre recombinase under the control of the dopamine transporter (DAT) promoter (DAT-Cre mice; Fig. 1a). In these mice, the expression of Cre is highly selective for DA neurons[34]. In line with this, we observed a high degree of overlap between Cre-dependent GCaMP6m expression and immunohistochemical staining against tyrosine hydroxylase (TH; Fig. 1c) in the SN. An optical fiber implanted in TS (Fig. 1a, b, Supplementary Fig. 1) enabled recording of Ca²⁺ transients in the axon terminals of DA neurons (Fig. 1d). In addition, to test whether the observed changes in fluorescence reflect neuronal activity we injected a Cre-dependent AAV expressing the control fluorophore EYFP in a separate group of control mice. Transient fluctuations in fluorescence were absent in mice expressing the control fluorophore (n = 2, Supplementary Fig. 2a, b), consistent with our previous results[25,28].

To examine DA terminal activity during associative fear learning, mice (n = 13) were trained in a FC paradigm (Fig. 1e) where a tone (CS) was paired with an aversive foot shock (US) on day 2, following a tone

habituation session (Hab) on day 1. On day 3, mice received a fear recall session consisting of CS presentations in the absence of the aversive US. During the course of FC, freezing to the CS gradually increased (Fig. 1f), indicating that the mice learned the association between the CS and the US. Mice showed significantly higher freezing levels to the CS at the end of FC session, as well as during fear recall, compared to Hab (Fig. 1f). Notably, the activity of DA terminals in TS appeared to resemble a PE signal during associative fear learning. In the beginning of FC, DA terminals in TS showed strong excitation to the aversive US which decreased during the course of conditioning (Fig. 1g–i), as the CS-US association was learned and the CS came to predict the occurrence of the US. Indeed, there was a significant decrease in US responses from the first to the last US (Fig. 1i). Conversely, while CS responses were absent at the beginning of FC, they gradually increased through the course of conditioning (Fig. 1g–i), mirroring the increase in freezing to the CS (Fig. 1f). In line with the behavioral results, responses to the CS were significantly larger during fear recall compared to Hab session (Fig. 1j), indicating that the CS responses were potentiated as the animals learned the CS-US association. In almost all animals, responses to the CS were larger during fear recall compared to Hab session (Fig. 1k). Notably, mice expressing a control fluorophore (EYFP) did not reveal any changes in fluorescence throughout the FC protocol (Supplementary Fig. 2e). Together, these results demonstrated that DA terminals in TS exhibited a PE-like activity pattern during associative fear learning.

Because the presentation of the aversive US itself could result in a nonspecific enhancement of CS responses, we next asked whether potentiation of CS responses during FC was indeed a result of associative learning and depended on the temporally contingent presentations of the CS and the US. To address this question, mice underwent an unpaired training paradigm where they received the same number of CS and US presentations as in FC, but the CS and US were explicitly unpaired in time during the training session on day 2 (Fig. 2a). In mice undergoing the unpaired training, we did not observe an increase in freezing to the CS between Hab, training and testing sessions (Fig. 2b). Mirroring these behavior results, we also did not find a significant difference in the CS response between the Hab and the testing session (n = 7; p = 0.31, signed-rank test Fig. 2c–e). These results indicate that potentiation of CS responses as well as the increased behavioral fear responses to the CS during FC required the association of the CS with the aversive US.

If DA terminals in TS encode a PE signal for the aversive US, we expect that responses to unpredicted USs should be larger in magnitude compared to responses to predicted USs. To test this, we examined DA terminal activity while previously well-trained mice received presentations of predicted (CS-US pairings) and unpredicted (US only) foot shocks (Fig. 2f). We indeed found stronger responses to unpredicted USs compared to CS-predicted ones (n = 9; p = 0.0078, signed-rank test; Fig. 2g, h), consistent with the decrease in US responses that we observed during the course of FC. Together, these results indicate that TS-projecting DA neurons signal a PE for aversive outcomes.

Our results show that TS-projecting DA neurons exhibit a positive prediction error for aversive stimuli. One question is whether these DA neurons exhibit negative prediction errors and respond to omission of aversive outcomes. To address this, we performed a partial conditioning task in which the aversive US was omitted randomly in half of the trials (Supplementary Fig. 3a). We found that DA terminals in TS did not exhibit a significant response to random omissions of the aversive US (Supplementary Fig. 3b, c), consistent with previous reports[35]. These results indicated a lack of inhibition by negative prediction errors in TS-projecting DA neurons.

Our results so far demonstrate that TS-projecting DA neurons are strongly activated by aversive USs. An important question is whether these DA neurons are activated also strongly by rewards. Since DA neurons are known to respond to rewards particularly when they are

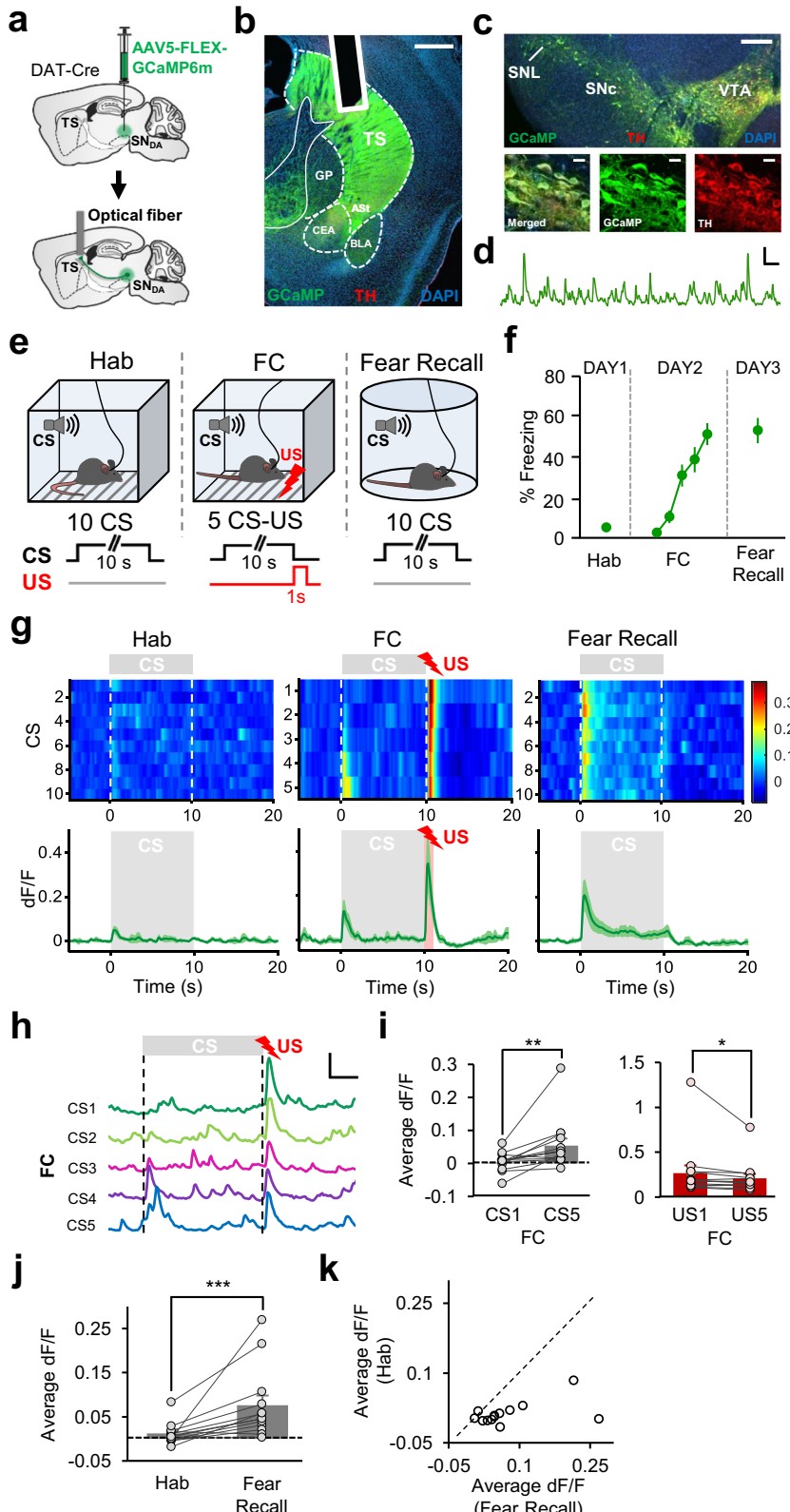

unpredicted[20–23,36], we used a reward task in which mice were trained to enter a reward port to receive rewards randomly with 50% probability (Fig. 2i), making reward delivery unpredicted. Interestingly, we found that DA terminals in TS exhibited weak responses to rewards (n = 19; Fig. 2j, k), consistent with previous findings[16,35]. In all animals tested, responses to unpredicted rewards were much smaller compared to unpredicted footshock USs (Fig. 2l). Furthermore, increasing the value

of reward did not have a significant effect on the magnitude of reward responses (Supplementary Fig. 4), suggesting that TS-projecting DA neurons do not encode the value of rewards, in line with previous reports[16]. Together, our results indicated that while DA terminals in TS were strongly activated particularly by painful and aversive stimuli, they responded only weakly to rewards, and did not respond to omission of aversive outcomes.

**Fig. 1 | DA terminals in TS encode a PE signal during associative fear learning.**
**a** Schematic of the surgical procedure. **b** Example histological image showing expression of GCaMP (green), tyrosine hydroxylase (TH, red) at DA terminals and DAPI (blue) staining in TS. The white vertical track indicates the optical fiber placement in TS. ASt amygdalostriatal transition area, BLA basolateral amygdala, CEA central nucleus of the amygdala, GP globus pallidus. Scale bar: 0.5 mm. **c** Top: example histological image. SNc substantia nigra pars compacta, SNL substantia nigra lateralis, VTA ventral tegmental area. Scale bar: 0.25 mm. Bottom: confocal images showing expression of GCaMP and TH staining and the merged image. Scale bar: 20 μm. **d** Example of change in fluorescence (dF/F) over time. Scale bar: 5 s, 0.5 dF/F. **e** Top: schematic of the behavioral protocol. Hab: tone habituation, FC: fear conditioning. Bottom: schematic of CS and US presentations. Schematic reprinted from ref. 28, copyright (2023), with permission from Elsevier. **f** Freezing to the CS ($n = 13$ mice) during Hab, FC and Fear Recall. **g** Top: average activity around each CS during Hab, FC and Fear Recall across all mice ($n = 13$). The heat map shows response amplitudes (dF/F). Bottom: average change in fluorescence around the time of CS (gray area). The red area during FC represents the US presentation. **h** Change in fluorescence around each CS and US during FC in an example animal. Scale bar: 2.5 s, 0.5 dF/F. **i** Left: comparison of average change in fluorescence during CS1 and CS5 of FC ($n = 13$, **$P = 0.0017$, two-sided signed-rank test). Right: comparison of average change in fluorescence during US1 and US5 of FC ($n = 13$, *$P = 0.013$, two-sided signed-rank test). **j** Average change in fluorescence during the CS for Hab and Fear Recall ($n = 13$, ***$P = 0.0007$, two-sided signed-rank test). **k** Scatter plot showing CS responses of each recording site during Hab and Fear Recall. The dashed line indicates the unity line. Data points below the unity line represent larger CS responses during Fear Recall. Shaded regions and error bars represent mean ± s.e.m. Source data are provided as a Source Data file.

## DA release in TS signals a PE during associative fear learning

DA neurons have been shown to co-release glutamate and GABA in the striatum[37,38]. It is therefore possible that the DA terminal activity might not reflect DA release during associative fear learning. To address this, we next examined the dynamics of DA release in TS during associative fear learning by performing optical recordings of the genetically encoded DA biosensor dLight[39] using fiber photometry. To this end, an AAV expressing dLight (AAV5-CAG-dLight1.1) was injected and optical fibers were implanted in the TS (Fig. 3a, b, Supplementary Fig. 5). Mice ($n = 8$) underwent the same fear conditioning protocol as in the GCaMP recording experiment (Fig. 3c), and showed increased freezing to the CS following FC (Fig. 3d), indicating successful fear learning.

In line with Ca²⁺ recordings in DA terminals, we found strong dLight responses to the US at the beginning of FC which significantly decreased through the course of conditioning (Fig. 3e, f), as occurrence of the US became predictable. Conversely, we observed a significant increase in the dLight activity in response to the CS when the first and last CSs of FC were compared (Fig. 3e, f), indicating potentiation of CS responses as the CS-US association was learned. Consistent with increased behavioral freezing, CS responses exhibited a significant increase from Hab to fear recall (Fig. 3e, g), and in all recording sites ($n = 10$) CS responses during fear recall were larger compared to Hab (Fig. 3g). Furthermore, we again observed small responses to unpredicted rewards (Fig. 3h–j), and in all dLight recording sites unpredicted footshock US responses (responses to the first US of FC) were larger compared to unpredicted reward responses (Fig. 3k). Taken together, these results indicate that DA release in TS underlies PE signaling during associative fear learning.

## TS-projecting DA neurons are required selectively for acquisition of cued associative fear learning

PE signals are thought to drive associative learning[19]. If DA neurons projecting to TS encode a PE signal and this signal is critical for driving associative fear learning, we expect that lesioning these DA neurons should impair particularly the acquisition of fear conditioning. To address this, we performed projection-specific ablation of TS-projecting DA neurons using a DA neuron selective neurotoxin, 6-hydroxydopamine (6-OHDA; Fig. 4a). Importantly, the 6-OHDA injections in TS caused reduction of DA axons specifically in TS, and not in the neighboring structures such as the amygdala and the amygdalostriatal transition area (ASt; Fig. 4b). Following 6-OHDA lesions, mice were trained using an auditory FC protocol (Fig. 4c) consisting of 5 CS-US pairings. Twenty-four hours after conditioning, mice underwent a fear recall test consisting of 5 CS presentations. We found that the lesioned mice ($n = 10$) froze significantly less to the CS throughout the conditioning session compared to saline-injected control mice ($n = 12$; Fig. 4d), suggesting impaired fear learning. Furthermore, impaired fear acquisition resulted in a weaker fear memory when tested the next day (Fig. 4d). 6-OHDA lesioned mice froze significantly less at the end of FC and the beginning of fear recall sessions

compared to control mice (Fig. 4e). These results demonstrated that TS-projecting DA neurons are required for the acquisition of the CS-US association.

A PE signal is expected specifically to be critical for initiating and driving new associative fear learning, but not the retrieval of fear memories. We therefore hypothesized that TS-projecting DA neurons might be critical selectively during fear acquisition on the conditioning day, but not later once the CS-US association was learned. To test this, we performed 6-OHDA ablation of TS-projecting DA neurons after fear memory was formed (Fig. 4f). In support of our hypothesis, we found that the control ($n = 8$) and the lesioned mice ($n = 8$) exhibited comparable levels of freezing to the CS during the fear recall test performed 3 weeks after 6-OHDA lesions (Fig. 4g, h), suggesting that the activity of these DA neurons was not necessary for retrieval and expression of fear memories. These results indicated that TS-projecting DA neurons were indeed important selectively for the acquisition of fear conditioning but were no longer required once the CS-US association was learned.

Given that TS is a sensory striatal region receiving sensory inputs such as auditory and visual[13], we hypothesized that DA projections to TS might be important for associating specifically discrete sensory cues with aversive outcomes. It is well-established that cued versus contextual fear conditioning involves distinct neural circuits[40]. We therefore next investigated whether TS-projecting DA neurons were necessary for contextual fear learning. To this end, we performed a contextual FC paradigm consisting of 5 US presentations in context A. Mice received 6-OHDA or saline injections in TS as described above and 3 weeks later underwent contextual FC (Fig. 4i). The animals were tested for contextual fear memory the next day (Fig. 4j). Lesioned ($n = 7$) and control ($n = 7$) mice showed comparable levels of freezing to the context during contextual testing (Fig. 4k), indicating that DA projections to TS were not required for learning the association between the context and the US. Together, these results suggest that TS-projecting DA neurons are critical for cued but not contextual fear learning. However, whether they are required for associating stimuli from sensory modalities other than auditory remained an important question. To address this, we conducted a visual FC paradigm using a discrete light cue as the CS. Mice received 6-OHDA or saline injections in TS as described above and 3 weeks later underwent visual FC (Supplementary Fig. 6a, b). We found that the lesioned mice ($n = 7$) froze significantly less to the CS throughout the conditioning session compared to saline-injected control mice ($n = 6$; Supplementary Fig. 6c, d), suggesting impaired visual fear learning. Furthermore, impaired fear acquisition resulted in a weaker fear memory when tested the next day (Supplementary Fig. 6c, d). These results demonstrated that TS-projecting DA neurons are necessary for cued associative fear learning.

Since we observed only small responses to rewards, we hypothesized that TS-projecting DA neurons might not be required for learning the association between a cue and reward. To test this, we

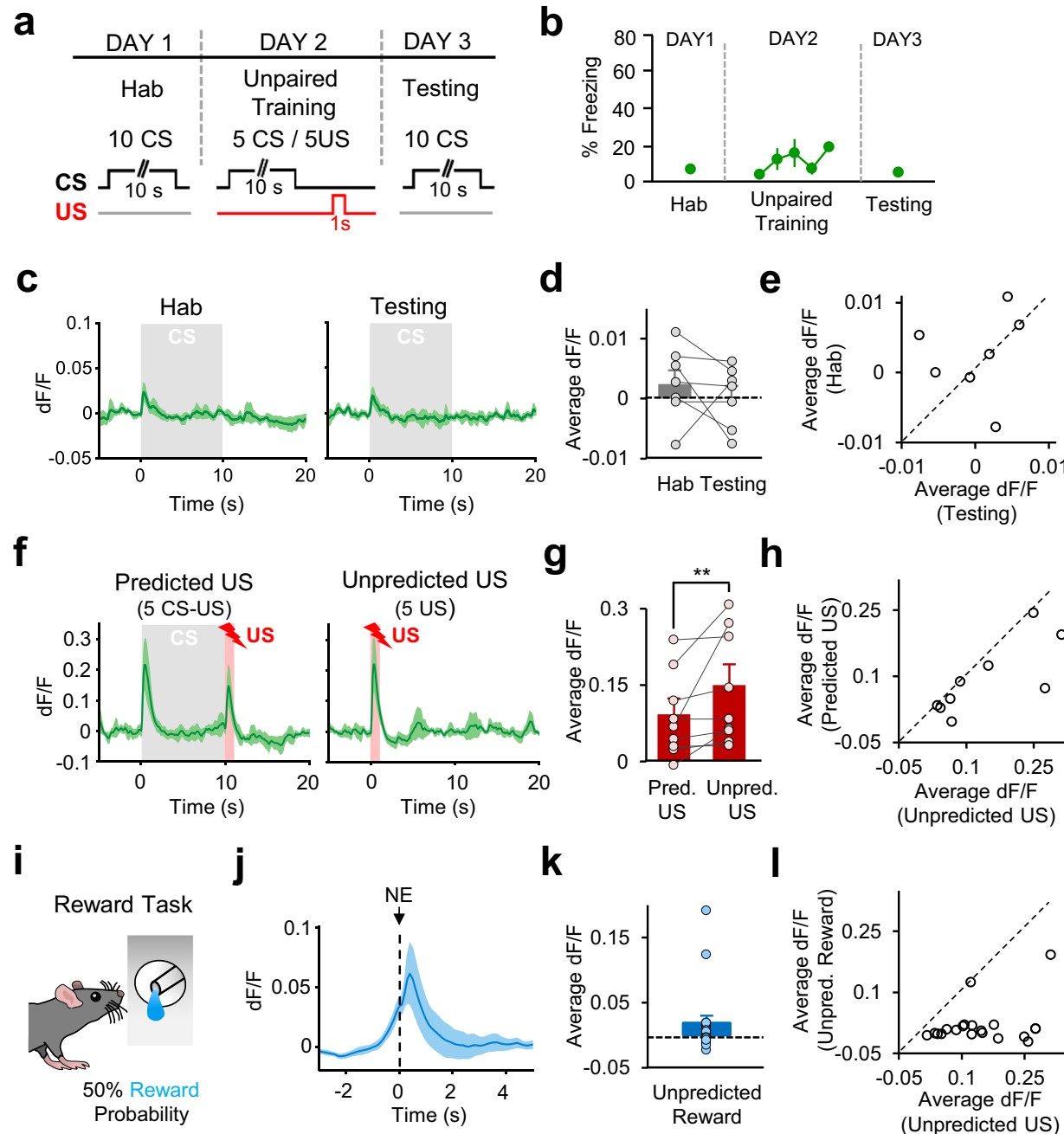

**Fig. 2 | Activity of DA terminals in TS in response to CSs, aversive USs, and rewards that are unpredicted. a** Top schematic of the behavioral protocol for unpaired training. Hab tone habituation. Bottom: Schematic of CS and US presentations during Hab, Unpaired Training, and Testing sessions. **b** Behavioral freezing to the CS (*n* = 6 mice) during Hab (average of 10 CSs), Unpaired Training and Testing sessions (average of 10 CSs). **c** Average change in fluorescence from recording sites in TS (*n* = 7) around the time of CS (gray area) during Hab, and Testing. **d** Comparison of average change in fluorescence in the 5 s after CS onset during Hab and Testing (*n* = 7, P = 0.31, two-sided signed-rank test). **e** Scatter plot showing CS responses of each recording site during Hab and Testing. The dashed line indicates the unity line. Data points below the unity line represent larger CS responses during Testing. **f** Average change in fluorescence from recording sites in TS (n = 9) around the time of CS (gray area) and US (red area) during Predicted and Unpredicted US presentations. **g** Comparison of average change in fluorescence

during US presentation for Predicted and Unpredicted USs (*n* = 9, **P = 0.0078, two-sided signed-rank test). **h** Scatter plot showing US responses of each recording site during Predicted and Unpredicted US presentations. The dashed line indicates the unity line. Data points below the unity line represent larger responses for the Unpredicted US. **i** Schematic of the reward task. Animals received reward randomly 50% of the time after entering the noseport. Schematic reprinted from ref. 28, copyright (2023), with permission from Elsevier. **j** Average change in fluorescence during rewarded noseport entries (NE) from all recording sites (*n* = 19). **k** Average change in fluorescence in the 3 s after noseport entry during rewarded NE (*n* = 19). **l** Scatter plot showing responses from each recording site (*n* = 19) during Unpredicted US and reward presentations. The dashed line indicates the unity line. Data points below the unity line represent larger responses for Unpredicted US. Shaded regions and error bars represent mean ± s.e.m. Source data are provided as a Source Data file.

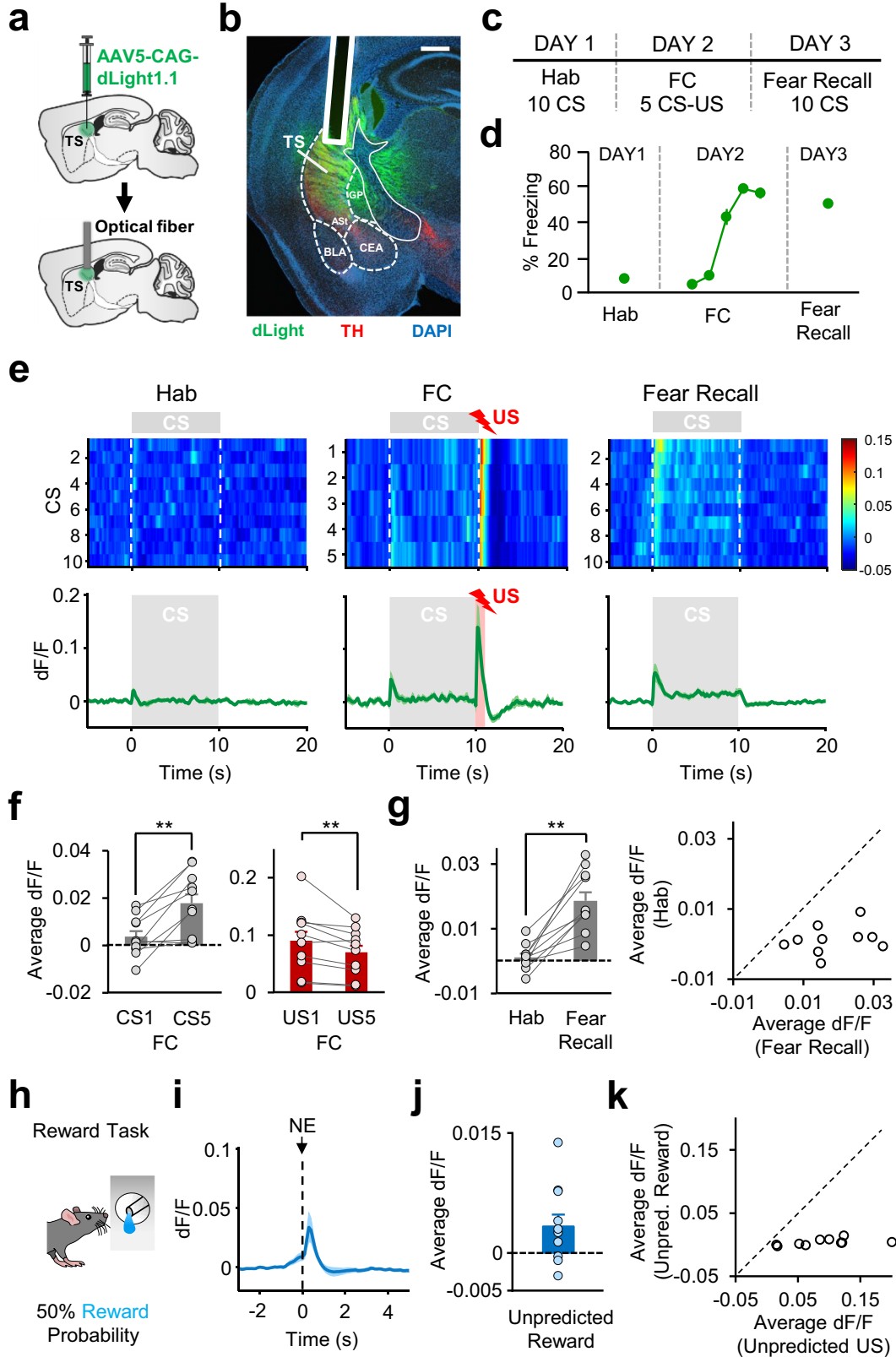

performed a reward learning task (Fig. 4l, m) in which a tone CS was paired with reward. The animals underwent the reward task following 6-OHDA lesioning of TS-projecting DA neurons (Fig. 4l). Notably, both lesioned and control groups showed similar learning rates during the reward task. The two groups did not differ in the number of rewarded CSs (Fig. 4n, o), latency to enter the port (Supplementary Fig. 7c, d), time they spent in the port during the CS (Supplementary Fig. 7e, f) as

well as the number of nose pokes during the ITIs (two-way repeated measures ANOVA, no main effect of group: $F_{1,52} = 1.6$, $P = 0.22$; no group × trial interaction: $F_{4,52} = 0.68$, $P = 0.6$). These results suggest that TS-projecting DA neurons are not required for associating cues with rewards and hence are not critical for associative reward learning.

Taken together, these results reveal a highly selective role for TS-projecting DA neurons in associative learning. We demonstrate that

**Fig. 3 | DA release dynamics in TS during associative fear learning and reward task. a** Schematic of the surgical procedure. **b** Example histological image showing expression of dLight (green), tyrosine hydroxylase (TH, red) and DAPI (blue) staining. White vertical track indicates the optical fiber placement. ASt amygdalostriatal transition area, BLA basolateral amygdala, CEA central nucleus of the amygdala, GP globus pallidus. Scale bar: 0.5 mm. **c** Schematic of the behavioral protocol. **d** Freezing to the CS ($n = 8$ mice) during Hab, FC and Fear Recall. **e** Top: Average dLight activity around each CS during Hab, FC and Fear Recall across all recording sites ($n = 10$). The heat map shows response amplitudes (dF/F). Bottom: average change in fluorescence around the time of CS presentation (gray area). The red area during FC represents the US presentation. **f** Left: comparison of average change in fluorescence during CS1 and CS5 of FC ($n = 10$, **$P = 0.008$, two-sided signed-rank test). Right: comparison of average change in fluorescence during US1

and US5 of FC ($n = 10$, **$P = 0.009$, two-sided signed-rank test). **g** Left: average change in fluorescence in the 5 s after CS onset during Hab and Fear Recall (**$P = 0.002$, two-sided signed-rank test). Right: scatter plot showing CS responses of each recording site ($n = 10$) during Hab and Fear Recall. Data points below the unity line represent larger CS responses during Fear Recall. **h** Schematic of the reward task. Animals received reward 50% of the time after entering the noseport. Schematic reprinted from ref. 28, copyright (2023), with permission from Elsevier. **i** Average change in fluorescence during rewarded noseport entries (NE, $n = 10$). **j** Average change in fluorescence in the 3 s after noseport entry during rewarded NE ($n = 10$). **k** Scatter plot showing responses from each recording site ($n = 10$) during unpredicted US (first US of FC) and unpredicted reward. Data points below the unity line represent larger responses for Unpredicted US. Shaded regions and error bars represent mean ± s.e.m. Source data are provided as a Source Data file.

they are requird selectively during the acquisition of fear learning, but not once the CS-US association is learned. Importantly, we found that these DA neurons are specifically important for learning discrete CS−US, but not context−US or CS−reward, associations.

## Temporally-precise activation of DA terminals in TS enhances associative fear learning

If DA activity in TS at the time of the US acts as a teaching signal and causes learning about the CS, then boosting this signal should enhance associative fear learning. To test this, we optogenetically excited DA terminals in TS precisely at the time of the US during FC. DAT-Cre mice were bilaterally injected with a Cre-dependent AAV expressing either channelrhodopsin-2 (ChR2) fused with EYFP (ChR2-EYFP) or EYFP only (EYFP control) in the SN, and implanted bilaterally with optical fibers above TS (Fig. 5a–c, Supplementary Fig. 8). We again observed a high level of overlap between Cre-dependent ChR2-EYFP expression and immunohistochemical staining against TH (Fig. 5c) suggesting DA neuron-specific expression of ChR2.

In order to examine enhancement of fear learning, mice were trained in a weak fear conditioning protocol (Fig. 5d) using a low US intensity (0.35 mA). The experimental group consisted of ChR2-EYFP expressing mice which received blue light stimulation specifically at the time of the US (US Paired-ChR2, $n = 12$; Fig. 5e). The control group expressing EYFP received the identical light delivery (US Paired-EYFP, $n = 8$). Comparison of freezing levels to the CS in the two groups revealed a significant difference between the ChR2 and the control mice during both FC and fear recall (Fig. 5i, j). The ChR2 group exhibited higher freezing levels to the CS compared to control mice suggesting enhanced fear conditioning. However, it is also possible that excitation of DA terminals in TS is aversive per se and pairing of the CS with this aversive outcome results in learning of the association between the CS and DA terminal excitation rather than enhancing the CS-footshock US association. To address this, we performed excitation of DA terminals in TS in the absence of the US (No-US Control; Fig. 5f) and found that this did not induce fear learning (Fig. 5i, j), suggesting that excitation of DA terminals in TS did not act as an aversive stimulus or a threat per se. Together, these results demonstrate that excitation of DA terminals in TS at the time of the US enhances associative fear learning.

We next asked whether DA activity in TS during the CS was critical for driving associative fear learning. To address this, we optogenetically excited DA terminals in TS during the CS presentations of FC (Fig. 5g). The ChR2 ($n = 7$) and EYFP ($n = 6$) expressing mice underwent the same FC protocol but this time received light stimulation specifically during the CS presentations (CS-paired). We found a significant difference between the two groups during both FC and fear recall test (Fig. 5k, l). Notably, light stimulation during the inter-trial intervals (ITI Control, Fig. 5h) did not have a significant effect on freezing levels and the ChR2-expressing mice froze more to the CS compared to EYFP and ITI controls (Fig. 5k, l), indicating that excitation of DA terminals in TS during the CS enhances associative fear learning.

However, it is also possible that excitation of DA neuron terminals in TS per se could affect movement, increase anxiety levels or induce aversive responses, potentially leading to a general increase in freezing rather than specifically enhancing the associative learning process. To address this, we performed real-time place preference, open field and elevated plus maze tests (Fig. 5m–o) to examine the effect of DA terminal excitation on avoidance, movement and anxiety-like behaviors. Exciting DA terminals in TS (ChR2-EYFP mice $n = 8$, EYFP mice $n = 8$) did not cause real-time place avoidance (Fig. 5m) suggesting that DA terminal excitation per se was not aversive. Furthermore, we also did not find a significant difference between the two groups in their anxiety-like behaviors when DA terminals in TS were illuminated in the open field (Fig. 5n, two-way repeated measures ANOVA, no main effect of group: $F_{1,28} = 0.18$, $P = 0.67$) and the elevated plus maze (Fig. 5o, two-way repeated measures ANOVA, no main effect of group, $F_{1,28} = 1.88$, $P = 0.19$). Finally, excitation of DA terminals in TS also did not have an effect on the animal's velocity in the open field (Fig. 5n, two-way repeated measures ANOVA, no main effect of group, $F_{1,28} = 1.03$, $P = 0.32$). Together, these results suggest that the observed effect on fear learning cannot simply be due to aversion, increased anxiety or changes in movement that was caused by the excitation of DA terminals in TS.

## Temporally-precise inhibition of DA terminals in TS impairs associative fear learning

If DA input to TS at the time of the US acts as a teaching signal and drives the learning about the CS, then inhibiting this signal is expected to impair associative fear learning. To address this, we optogenetically inhibited DA terminals in TS precisely at the time of the US during FC. DAT-Cre mice were bilaterally injected with a Cre-dependent AAV expressing either archaerhodopsin (eArch) fused with EYFP (eArch-EYFP) or EYFP only (EYFP control) in the SN, and implanted bilaterally with optical fibers above TS (Fig. 6a–c, Supplementary Fig. 9). We again observed a high level of overlap between Cre-dependent eArch-EYFP expression and immunohistochemical staining against TH (Fig. 6c) suggesting DA neuron-specific expression of eArch.

The experimental group consisted of eArch-EYFP expressing mice which received yellow light delivery specifically at the time of the US ($n = 7$; Fig. 6d). The control group expressing EYFP received the identical light delivery ($n = 6$). Comparison of freezing levels to the CS in the two groups revealed a significant difference between the eArch and the control mice during both FC and fear recall (Fig. 6e, f). The eArch group exhibited lower freezing levels to the CS compared to control mice suggesting impaired fear conditioning. These results indicate that activation of DA terminals in TS at the time of the US is required for associative fear learning.

To test whether inhibition of DA neuron terminals in TS per se could affect movement, decrease anxiety levels or have a rewarding effect, potentially leading to a general decrease in freezing rather than specifically impairing the associative learning process, we performed real-time place preference, open field and elevated plus maze tests

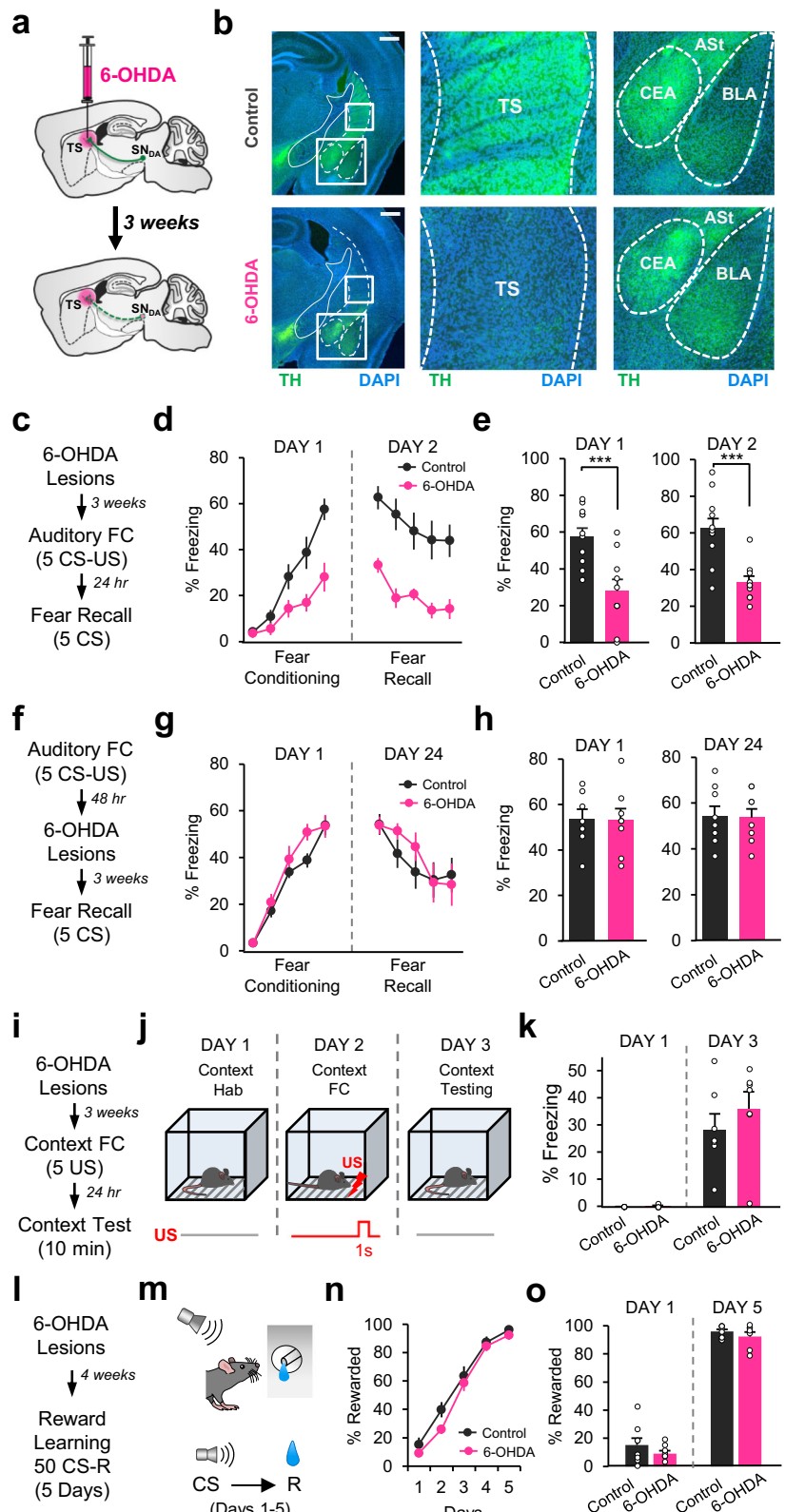

(Fig. 6g–i). Inhibiting DA terminals in TS (eArch-EYFP mice $n = 7$, EYFP mice $n = 6$) did not cause real-time place preference (Fig. 6g) suggesting that DA terminal inhibition per se did not have a rewarding effect. Furthermore, we also did not find a significant difference between the two groups in their anxiety-like behaviors when DA terminals in TS were inhibited in the open field (Fig. 6h; two-way repeated measures ANOVA, no main effect of group: $F_{1,22} = 1.57$, $P = 0.23$; no group × trial interaction: $F_{2,22} = 1.57$, $p = 0.22$) and the elevated plus maze (Fig. 6i; two-way repeated measures ANOVA, no main effect of group: $F_{1,22} = 0.19$, $P = 0.67$). Finally, inhibition of DA terminals in TS also did not have an effect on the animal's velocity in the open field (Fig. 6h; two-way repeated measures ANOVA, no main effect of group: $F_{1,22} = 0.10$, $P = 0.75$; no group × trial interaction: $F_{2,22} = 1.51$, $p = 0.24$). Together, these results indicate that the observed

**Fig. 4 | TS-projecting DA neurons are required selectively for the acquisition of cued associative fear learning. a** Schematic of the surgical procedure. **b** Example sections showing tyrosine hydroxylase (TH, green) and DAPI (blue) staining in saline (top, left) and 6-OHDA (bottom, left) groups. Scale bar: 0.5 mm. Close-up images showing TH-staining in TS (middle) and the amygdala (right; ASt, amygdalostriatal transition area; BLA, basolateral amygdala; CEA, central amygdala) in saline (top) and 6-OHDA (bottom) groups. **c** Schematic of the protocol. **d** Freezing to the CS for saline ($n = 12$) and 6-OHDA ($n = 10$) groups (two-way repeated measures ANOVA, FC: main effect of group: $F_{1,80} = 7.84$, $P = 0.011$; fear recall: main effect of group: $F_{1,80} = 18.60$, $P = 0.0003$). **e** Left: freezing to the last CS during FC (two-sided t-test, t(20) = 3.85, $P = 0.001$). Right: freezing to the first CS of fear recall (two-sided t-test, t(20) = 4.68, $P = 0.0001$). **f** Schematic of the protocol. **g** Freezing to the CS for saline ($n = 8$) and 6-OHDA ($n = 8$) groups (two-way repeated measures

ANOVA, FC: no main effect of group: $F_{1,56} = 1.37$, $P = 0.26$; fear recall: no main effect of group: $F_{1,56} = 0.12$, $P = 0.72$). **h** Freezing to the last CS during FC (left, two-sided t-test, t(14) = 0.06, $P = 0.95$) and to the first CS during fear recall (right, two-sided t-test, t(14) = 0.07, $P = 0.94$). **i** Schematic of the protocol. **j** Schematic of the task. Schematic reprinted from ref. 28, copyright (2023), with permission from Elsevier. **k** Freezing to the context in control ($n = 7$) and 6-OHDA ($n = 7$) groups. (two-sided t-tests: Context Hab, $P = 0.14$; Context Test, $P = 0.38$). **l** Schematic of the protocol. **m** Schematic of the task. Schematic reprinted from ref. 28, copyright (2023), with permission from Elsevier. **n** Percent rewarded CSs for saline ($n = 8$) and 6-OHDA ($n = 7$) groups (two-way repeated measures ANOVA, no main effect of group: $F_{1,52} = 1.7$, $P = 0.21$). **o** Percent rewarded CSs during the first and last days (saline, $n = 8$; 6-OHDA, n = 7). Error bars represent mean ± s.e.m. Source data are provided as a Source Data file.

---

impairment in fear learning was due to inhibition of DA terminals in TS, rather than nonspecific effects.

### Neuronal activity in TS exhibits a PE-like pattern and potentiation of CS responses during associative fear learning

The requirement of a DA PE signal in TS during FC suggests that the activity of TS neurons is likely critical for acquisition of associative fear learning. To address this, we first examined the neuronal activity in TS during FC by measuring activity-dependent $Ca^{+2}$ signals using fiber photometry. To this end, an AAV expressing the $Ca^{+2}$ indicator GCaMP6f was injected and an optical fiber was implanted in the TS (Fig. 7a, b, Supplementary Fig. 10). To test whether the observed changes in fluorescence depend on neuronal activity we injected an AAV expressing the control fluorophore EYFP in a separate group of control mice. Transient fluctuations in fluorescence were absent in mice expressing the control fluorophore ($n = 2$, Supplementary Fig. 2c, d). The GCaMP-expressing mice ($n = 9$) underwent the same FC protocol as in fiber photometry experiments described above (Fig. 7c), and exhibited successful fear learning (Fig. 7d). Interestingly, we observed that the activity in TS exhibited a PE-like pattern during FC (Fig. 7e), similar to the results of our recordings of DA terminal activity as well as DA release in TS. While the responses to the US decreased during the course of FC, the CS responses gradually increased (Fig. 7e). Indeed, there was a significant increase in responses to the CS from first to last CSs (Fig. 7f). Conversely, the responses to the US were larger to the first compared to the last US (Fig. 7f). Importantly, we found a significant increase in responses to the CS from Hab to fear recall (Fig. 7g). In almost all recording sites, the CS responses during fear recall were larger in amplitude compared to Hab (Fig. 7g). Together, these results demonstrated a PE-like activity pattern in TS during associative fear learning.

### Causal Contribution of Neuronal Activity in TS to Associative Fear Learning

We next asked whether activity in TS was necessary for associative fear learning. To address this, we first performed chemogenetic inhibition of TS neurons during FC. Mice received injections of an AAV expressing the inhibitory DREADD receptor (hM4D(Gi)) in TS (Fig. 8a, b) and underwent the same auditory FC protocol as in the 6-OHDA experiment (Fig. 8c). Thirty minutes before the FC session, mice received systemic injections of the DREADD agonist clozapine N-oxide (CNO) to inhibit activity of TS neurons during fear conditioning whereas control mice received saline injections. We found that CNO-injected mice ($n = 8$) froze significantly less to the CS at the end of FC compared to saline-injected controls ($n = 8$; Fig. 8d, e) suggesting impaired fear learning. Furthermore, impaired fear learning resulted in a weaker fear memory when tested the next day during the fear recall test (Fig. 8d, f). These effects were dependent on inhibitory DREADD receptor expression since CNO injection had no effect in mCherry-expressing control mice (Supplementary

Fig. 11). These results demonstrated that the activity of TS neurons is critical for associative fear learning.

A PE-like activity pattern in TS for the aversive US suggests that this signal might drive fear learning. If that is the case, the activity of TS neurons during the aversive US is expected to act as a teaching signal for associative fear learning. We therefore next investigated whether TS neuronal activity during the US is necessary for driving fear learning. To this end, we performed temporally-precise optogenetic inhibition of TS neurons at the time of the US during FC (Fig. 8i). An AAV expressing eArch-EYFP or EYFP only was bilaterally injected and optic fibers were bilaterally implanted in TS (Fig. 8g, h, Supplementary Fig. 12). We found a significant difference between the eArch ($n = 7$) and the control ($n = 8$) mice during FC (Fig. 8j). eArch-expressing mice showed significantly lower freezing to the CS at the end of FC (Fig. 8k) and the beginning of fear recall test (Fig. 8l), suggesting impaired fear learning. These results indicate that the activity of TS neurons at the time of the US acts as a teaching signal and is necessary for driving associative fear learning.

If activity of TS neurons during the US drives associative fear learning, then boosting this activity is expected to enhance fear conditioning. We tested this by performing temporally-precise optogenetic excitation of TS neurons at the time of the US during FC (Fig. 8o). To this end, an AAV expressing ChR2-EYFP or EYFP only was bilaterally injected and optic fibers were bilaterally implanted in TS (Fig. 8m, n, Supplementary Fig. 13). In order to examine enhancement of fear learning, mice were trained in a weak fear conditioning protocol (Fig. 8o), as during optogenetic excitation of DA terminals described above (Fig. 5). We found a significant difference between the ChR2 ($n = 8$) and the control ($n = 8$) mice during FC (Fig. 8p). ChR2-expressing mice showed significantly higher freezing to the CS at the end of FC (Fig. 8r) and the beginning of fear recall test (Fig. 8s), indicating that excitation of TS neurons at the time of the US enhances associative fear learning.

To test whether manipulation of TS neuronal activity per se could affect movement, decrease anxiety levels or could have an aversive effect, potentially leading to a general change in freezing rather than specifically affecting the associative learning process, we tested animals on the control tasks while inhibiting (Supplementary Fig. 14a, b) or exciting (Supplementary Fig. 14c-e) the activity of TS neurons. Chemogenetic inhibition of TS neurons (CNO group $n = 6$, saline group $n = 6$) did not cause anxiety-like behaviors in the open field and the elevated plus maze (Supplementary Fig. 14a, b). Inhibition of TS neurons also did not have an effect on the animal's velocity in the open field (Supplementary Fig. 14a). Conversely, exciting TS neurons did not cause real-time place avoidance (Supplementary Fig. 14c), anxiety-like behaviors in the open field (Supplementary Fig. 14d) or the elevated plus maze (Supplementary Fig. 14e) and did not affect velocity in the open field (Supplementary Fig. 14d). Together, these results indicate that the observed impairment and enhancement in fear learning was due to inhibition and excitation of TS neurons, respectively, rather than nonspecific effects.

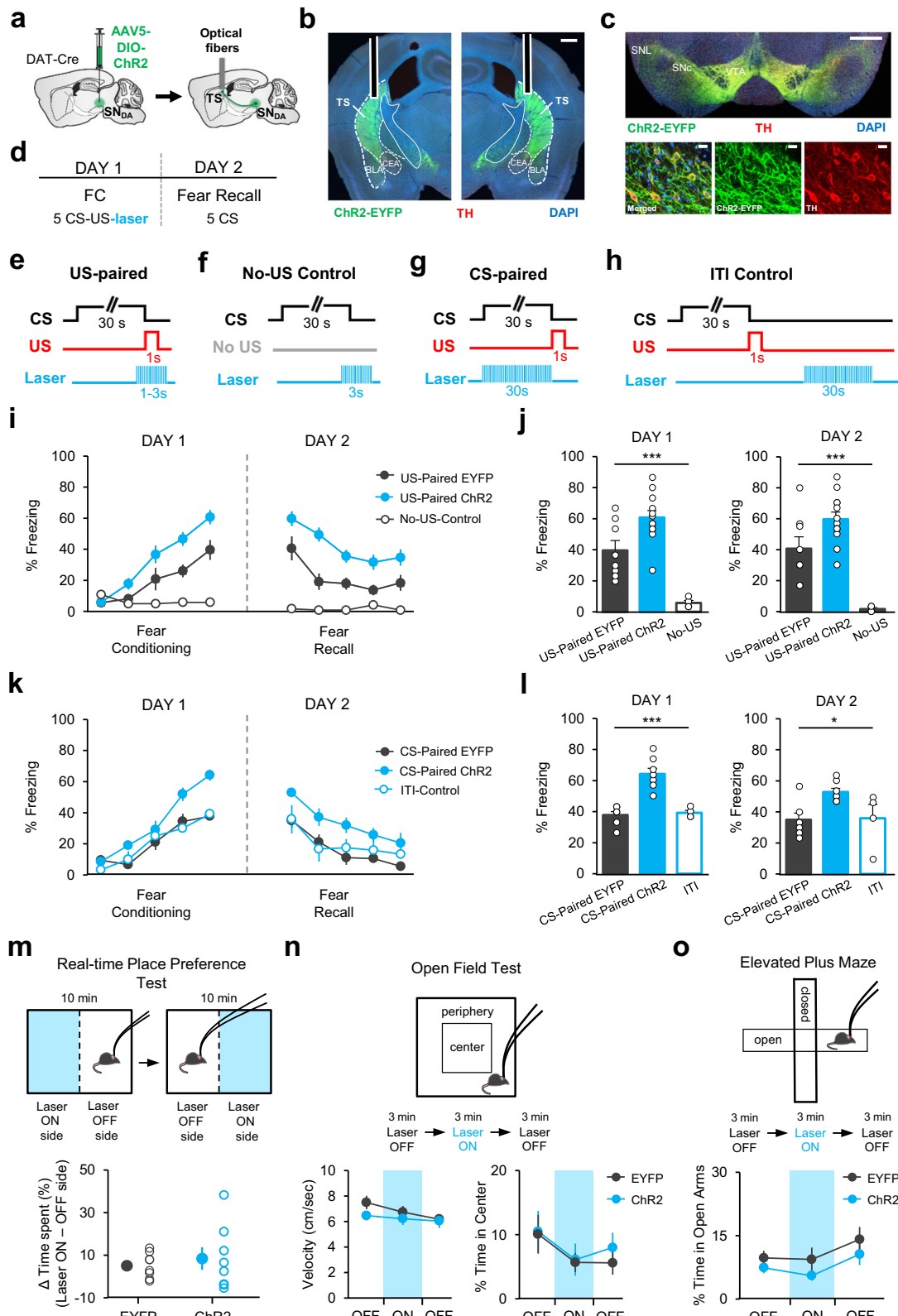

## DA Input is Critical for PE-like Activity and Potentiation of CS Responses in TS during Associative Fear Learning

Our results demonstrated that the activity in TS exhibits a PE-like pattern during fear learning. A key question is whether DA input is important for this fear learning-related activity observed in TS. To address this question, we performed 6-OHDA ablation of TS-projecting DA neurons followed by Ca²⁺ recordings in TS using

fiber photometry. Mice received 6-OHDA or saline injections, and one week later, were injected with an AAV expressing GCaMP6f as well as implanted with an optical fiber in the TS (Fig. 9a, Supplementary Fig. 15). The animals underwent the same FC protocol (Fig. 9b) as in fiber photometry experiments described above. Importantly, 6-OHDA injected animals ($n = 8$) showed reduction in DAergic innervation of TS compared to control group ($n = 10$;

**Fig. 5 | Temporally-precise activation of DA terminals in TS enhances associative fear learning. a** Schematic of the surgery. **b** Example histological image showing expression of ChR2-EYFP (green), tyrosine hydroxylase (TH, red), and DAPI (blue) staining in TS. White vertical tracks indicate optical fiber placements. Scale bar: 0.5 mm. **c** Top: example image of the midbrain showing expression of ChR2-EYFP, TH, and DAPI staining. Scale bar: 0.5 mm. Bottom: confocal images. Scale bar: 20 μm. **d** Schematic of the behavioral protocol. Schematic of optogenetic excitation for US-Paired (**e**), No-US-Control (**f**), CS-Paired (**g**), and ITI-Control (**h**) groups. **i** Freezing to the CS in ChR2 ($n = 12$), EYFP ($n = 8$) and No-US-Control ($n = 4$) groups (two-way repeated-measures ANOVA, FC: main effect of group: $F_{2,84} = 12.97$, $P = 0.0002$; Fear recall: main effect of group: $F_{2,84} = 21.99$, $P < 0.0001$). **j** Left: freezing to the last-CS of FC (one-way ANOVA, $F_{2,23} = 19.78$, $P < 0.0001$). Right:

freezing to the first-CS of fear recall (one-way ANOVA, $F_{2,23} = 17.16$, $P < 0.0001$). **k** Freezing to the CS in ChR2 ($n = 7$), EYFP ($n = 6$) and ITI-Control ($n = 4$) groups (two-way repeated-measures ANOVA, FC: main effect of group: $F_{2,56} = 11.04$, $P = 0.0013$; fear recall: main effect of group: $F_{2,56} = 5.77$, $P = 0.014$). **l** Left: freezing to the last-CS of FC (one-way ANOVA, $F_{2,16} = 19.78$, $P < 0.0001$). Right: freezing to the first-CS of fear recall (one-way ANOVA, $F_{2,16} = 3.86$, $P = 0.04$). Top: schematic of the task. Bottom: (**m**), time spent in laser ON minus laser OFF side for EYFP ($n = 8$) and ChR2 ($n = 8$) groups during the real-time place preference test ($P = 0.56$, two-sided t-test). **n** Velocity (left) and time spent in the center of the open field (right) for EYFP ($n = 8$) and ChR2 ($n = 8$) groups. **o** time spent in the open arms of the elevated plus maze for EYFP ($n = 8$) and ChR2 ($n = 8$) groups. Error bars represent mean ± s.e.m. Source data are provided as a Source Data file.

Fig. 9d, h), and consistently, lesioned mice showed impaired fear learning compared to controls (Fig. 9c).

The PE-like activity pattern during FC that the control group exhibited ($n = 14$ recording sites; Fig. 9e–g) was not observed in the 6-OHDA lesioned group ($n = 14$ recording sites; Fig. 9i–k). In lesioned mice, we did not find an increase in the CS responses from first to last CSs (Fig. 9j), which was seen in control animals (Fig. 9f). Interestingly, the significant decrease in responses between the first and the last USs of FC, that was observed in control mice (Fig. 9f), was also absent in lesioned animals (Fig. 9j). Furthermore, while CS responses were significantly increased between Hab and fear recall in control animals, no increase was observed in lesioned mice (Fig. 9g, k). Whereas CS responses were larger in magnitude during fear recall compared to Hab in the majority of control animals, this was not the case in lesioned mice (Fig. 9g, k). Notably, there was no difference between the two groups in their CS responses during Hab, whereas during FC and fear recall, CS responses were significantly larger in the control group (Fig. 9l). The responses to unpredicted US (first US of FC) were not significantly different between the two groups (Fig. 9l), suggesting that US input to TS in general was not affected. Importantly, the difference in US responses from the first to the last US of FC was significantly different between the groups (Fig. 9m), indicating that the decrease in US responses as fear conditioning progressed was absent in 6-OHDA mice, consistent with a deficit in learning about the CS-US association in these mice. Overall, these results indicate that DA input is required for fear learning-induced activity patterns observed in TS during associative fear learning.

## Discussion

Whether midbrain DA neurons projecting to brain structures outside the canonical amygdala circuitry encode a PE signal that is critical to drive associative fear learning is largely unknown. Recent studies have implicated TS in fear learning[11,12], yet the precise role DA neurons projecting to TS play in associative fear learning is incompletely understood. Here, we demonstrated that DA projections to TS drive associative fear learning by encoding a PE signal that is important for generating fear learning-induced activity patterns in TS. We first showed that DA projections to TS exhibit a positive PE signal during FC, and that this PE signal is transmitted by DA release in TS. Projection-specific lesioning of TS-projecting DA neurons selectively impaired the acquisition of associative fear learning, but not fear retrieval and expression. Notably, these neurons were required specifically for acquiring the association between the cue CS and US, but not between the context and US or between a CS and reward. Bidirectional temporally-precise optogenetic manipulations of DA projections to TS demonstrated the necessity of DA PE signaling in associative fear learning. Furthermore, activity in TS exhibited a PE-like pattern during FC. The neuronal activity in TS was required for fear learning, and temporally-precise enhancement of this activity enhanced fear learning. Finally, we demonstrated that DA input was critical for the fear learning-induced activity pattern in TS.

Previous studies using lesioning, pharmacological manipulations, recordings of DA neuronal activity, measurements of DA release as well as research on DA-deficient mice have consistently indicated a critical role of dopamine in fear learning and memory as well as aversion[30–33]. Specifically, while research on DA-deficient mice demonstrated the requirement of DA in fear learning[41], early pharmacological studies highlighted the role of DAergic signaling particularly in the basolateral amygdala (BLA) during acquisition and expression of fear memories[42–46]. Consistent with these findings, in DA-deficient mice, restoring DA production specifically in projections to BLA and striatum reversed deficits in fear memory formation[47]. Furthermore, disrupting phasic firing in dopamine neurons has been shown to impair fear learning[48,49]; and midbrain dopamine neurons exhibit phasic firing in response to aversive USs as well as CSs that predict them[16,25,29,35,49–56]. Interestingly, recent studies showed that DA terminals in BLA are activated by both aversive and rewarding stimuli, suggesting that these DA neurons encode the motivational salience of stimuli[57,58]. However, despite this extensive literature on the role of DA in fear learning and memory, it still remained elusive whether ventral midbrain dopamine neurons, located in the VTA and SN, encode prediction error signals necessary for driving associative fear learning.

The midbrain DA system is composed of functionally distinct and mostly non-overlapping subpopulations of DA neurons, each of which projects mainly to a specific brain region[15,59–63]. Notably, DA neurons projecting to TS have recently emerged as a unique subpopulation based on their distinct input-output circuitry[15]. These DA neurons have been shown to be activated by novel as well as external aversive stimuli (e.g. air puffs and loud tones but not bitter taste[16,35]). In line with these previous studies, we found that TS-projecting DA neurons are activated particularly strongly by aversive stimuli that are noxious such as foot shocks, but only weakly by rewards. Interestingly, TS-projecting DA neurons have been shown to exhibit a PE-like activity pattern during Pavlovian association of olfactory cues with mildly aversive stimuli such as air puffs[16,35]; and these DA neurons are particularly important for novelty-induced threat avoidance[16,17]. Although, recent studies showed enhancement of CS responses in TS-projecting DA neurons during fear conditioning[12], whether these neurons encode a PE signal for strong painful stimuli such as foot shocks; and whether this signal is necessary for associating cues with the aversive US during FC has remained elusive.

During associative learning, a positive PE acts as a teaching signal and drives learning about the CS that precedes the US[19,24,64–66]. We here demonstrate that TS-projecting DA neurons encode a positive PE for the aversive US during FC. These neurons respond more strongly to aversive USs when they are unpredicted compared to predicted ones. Furthermore, we observed that during the course of FC the US responses decreased whereas the CS responses were enhanced as the CS-US association was learned and the CS came to predict the US. These changes in the activity of TS-projecting DA neurons were specifically dependent on the temporally contingent presentation of the CS and the US: in mice trained with unpaired conditioning, during which the CS and the US were presented in a temporally unpaired

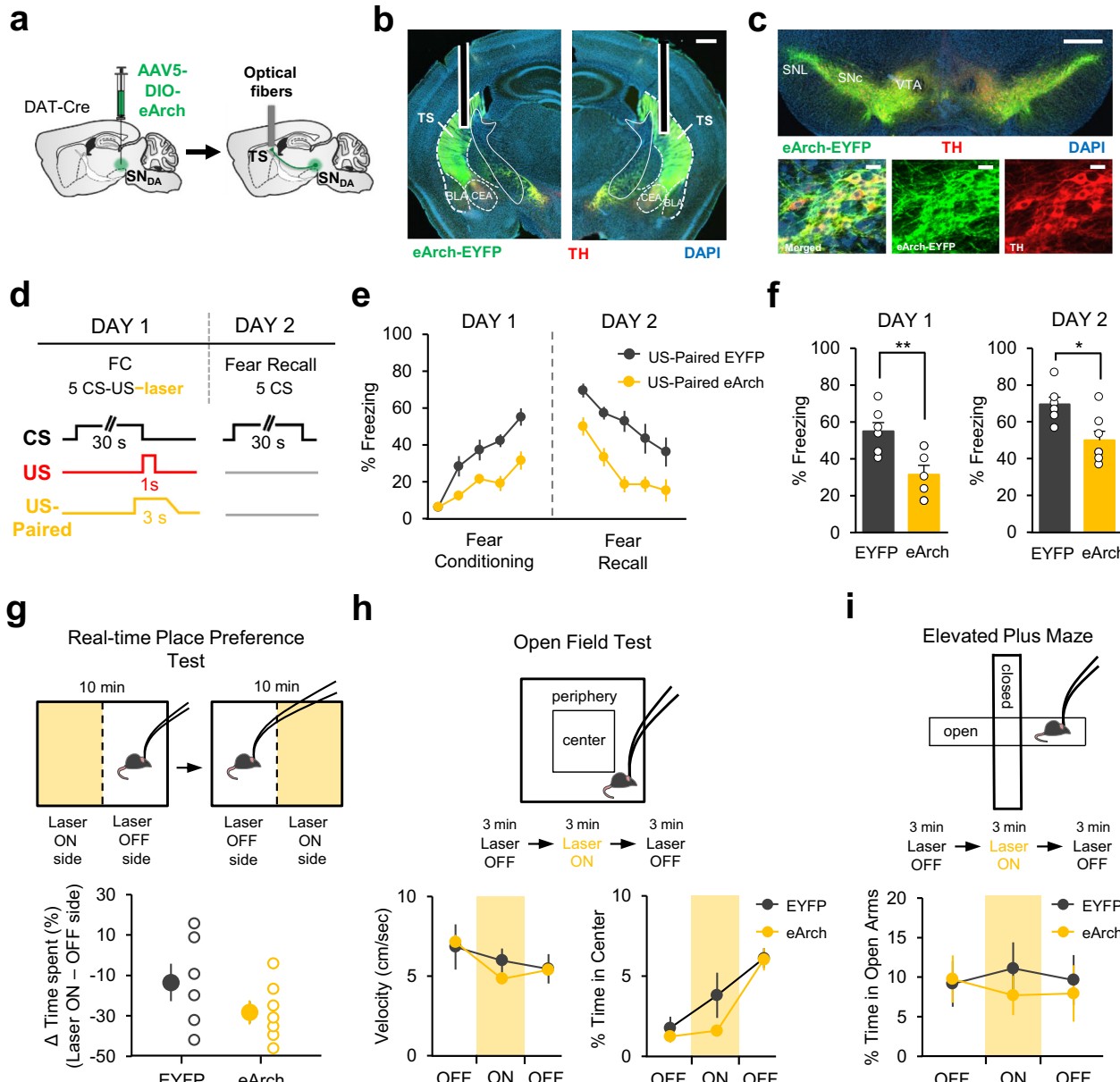

**Fig. 6 | Temporally-precise inhibition of DA terminals in TS impairs associative fear learning. a** Schematic of the surgery. **b** Example histological image showing expression of eArch-EYFP (green), tyrosine hydroxylase (TH, red), and DAPI (blue) staining in TS. White vertical tracks indicate the optical fiber placements in TS. Scale bar: 0.5 mm. **c** Top: example histological image showing expression of eArch-EYFP, TH and DAPI staining in the midbrain. Scale bar: 0.5 mm. Bottom: confocal images. Scale bar: 20 μm. **d** Top: schematic of the behavioral protocol. FC: fear conditioning. Bottom: schematic of the optogenetic inhibition paired to the US (US-Paired). **e** Freezing to the CS during FC and fear recall for US-Paired eArch ($n = 7$) and EYFP ($n = 6$) groups (two-way repeated measures ANOVA, FC: main effect of group: $F_{1,44} = 15.03$, $p = 0.0026$; fear recall: main effect of group: $F_{1,44} = 17.02$, $p = 0.0017$). **f** Left: freezing to the last CS of FC (two-sided t-test, t(11) = 3.3,

$p = 0.007$). Right: freezing to the first CS of fear recall (two-sided t-test, t(11) = 2.95, $p = 0.013$). **g** Top: schematic of the real-time place preference test. Bottom: difference between the percent of time mice spent in laser ON minus laser OFF side for EYFP ($n = 6$) and eArch ($n = 7$) groups (two-sided t-test, t(11) = 1.42, P = 0.18). **h** Top: schematic of the open field test. Bottom: velocity (left) and time spent in the center of the open field (right) during laser ON and OFF epochs for EYFP ($n = 6$) and eArch ($n = 7$) groups. Velocity and time in the center of the open field were comparable between the groups. **i** Top: schematic of the elevated plus maze test. Bottom: time mice spent in the open arms of the elevated plus maze during laser ON and OFF epochs for EYFP ($n = 6$) and eArch ($n = 7$) groups. Error bars represent mean ± s.e.m. across animals. Source data are provided as a Source Data file.

manner and thus CS did not predict the occurrence of the US, the CS responses failed to potentiate. Interestingly, these DA neurons have previously been shown to lack inhibitory responses to the omission of air puff USs[35]. Consistent with this, we also did not observe inhibition in response to random footshock US omissions, suggesting that TS-projecting DA neurons exhibit only positive PEs and lack negative PE signaling for aversive stimuli. Notably, inhibitory responses to aversive US omissions were also absent in the dorsal tegmental DA neurons that

have been shown to encode PE signals to gate fear learning[29], further supporting the notion of a general lack of negative PE signaling in DA neurons encoding PEs for aversive stimuli. Taken together, these findings suggest that DA prediction error signaling mechanisms may differ between rewarding versus aversive outcomes.

A PE signal is expected to initiate new learning; consistent with this, we demonstrate that ablation of TS-projecting DA neurons selectively impaired the acquisition of the CS-US association during

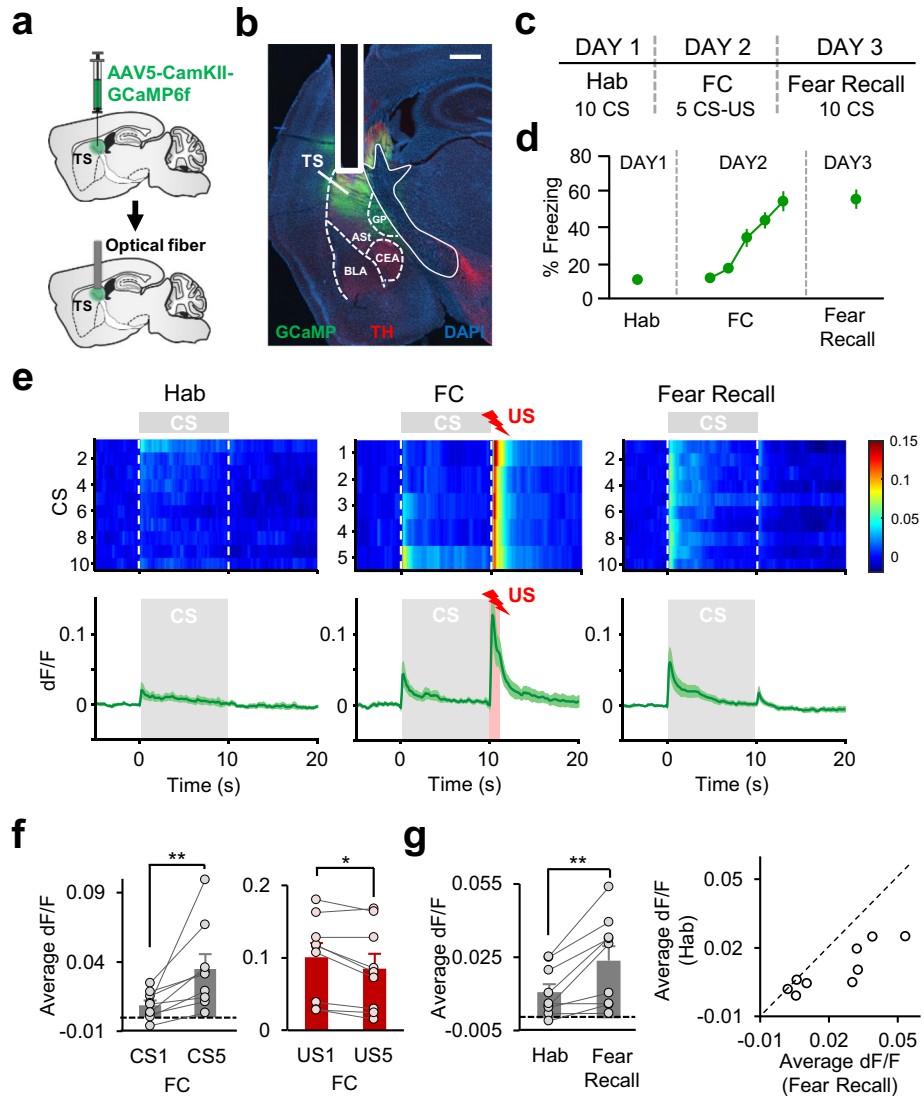

**Fig. 7 | Activity in TS exhibits a PE-like pattern, and CS responses are potentiated during associative fear learning. a** Schematic of the surgical procedure. **b** Example histological image showing expression of GCaMP (green) along with immunostaining for tyrosine hydroxylase (TH, red) and DAPI (blue) staining in TS. The white vertical track indicates the optical fiber placement in TS. ASt amygdalostriatal transition area, BLA basolateral amygdala, CEA central nucleus of the amygdala; GP globus pallidus. Scale bar: 0.5 mm. **c** Schematic of the behavioral protocol. Hab: tone habituation; FC: fear conditioning. **d** Behavioral freezing to the CS ($n = 9$ mice) during Hab, FC, and Fear Recall. **e** Top: average GCaMP activity around each CS during Hab, FC and Fear Recall across all recording sites ($n = 9$). The heat map shows response amplitudes (dF/F) around each CS presentation. Bottom: average change in fluorescence around the time of CS presentation (gray area) during Hab, FC, and Fear Recall. The red area during FC represents the US

presentation. **f** Left: comparison of average change in fluorescence in the 5 s after CS onset during CS1 and CS5 of FC. Note the significant increase in the Ca²⁺ signal from CS1 to CS5 ($n = 9$, **$P = 0.0039$, two-sided signed-rank test). Right: comparison of average change in fluorescence during the US (1 s) for US1 and US5 of FC. Note the significant decrease in the Ca²⁺ signal from US1 to US5 ($n = 9$, *$P = 0.02$, two-sided signed-rank test). **g** Left: average change in fluorescence in the 5 s after CS onset during Hab and Fear Recall. Note the significant increase in the Ca²⁺ signal from Hab to Fear Recall (**$P = 0.0078$, two-sided signed-rank test). Right: scatter plot showing CS responses of each recording site ($n = 9$) during Hab and Fear Recall. Data points below the unity line represent larger CS responses during Fear Recall. Shaded regions and error bars represent mean ± s.e.m. Source data are provided as a Source Data file.

FC, but not retrieval and expression of fear memories. In line with our results, ablation of these DA neurons has previously been shown to impair learning about and avoiding aversive air puffs[16]. Interestingly, we found that TS-projecting DA neurons were not required for acquisition of contextual fear memories, highlighting the selective role of these DA neurons in cued associative fear learning. Furthermore, we demonstrate that exciting DA projections to TS at the time of the CS or the US caused enhancement of FC. Conversely, optogenetic inhibition of these DA projections during the US impaired FC. However, in a recent study optogenetic inhibition of these DA projections during fear conditioning affected retrieval but not the acquisition of fear memories[12]. The reasons for these varying results are not clear.

Nevertheless, our results indicate that the PE signal encoded by TS-projecting DA neurons for aversive stimuli indeed acts as a teaching signal to drive the acquisition of CS-US association during fear learning.

In contrast to their strong responses to the aversive US, we show that DA projections to TS are only weakly activated by rewards. Consistent with this, we found that TS-projecting DA neurons were not required for associating an auditory cue with reward; although they have been implicated in more complex forms of reward learning involving auditory discrimination[67]. It is important to note that although the operant reward task that we used is not fully comparable with classical conditioning, our results nevertheless indicate

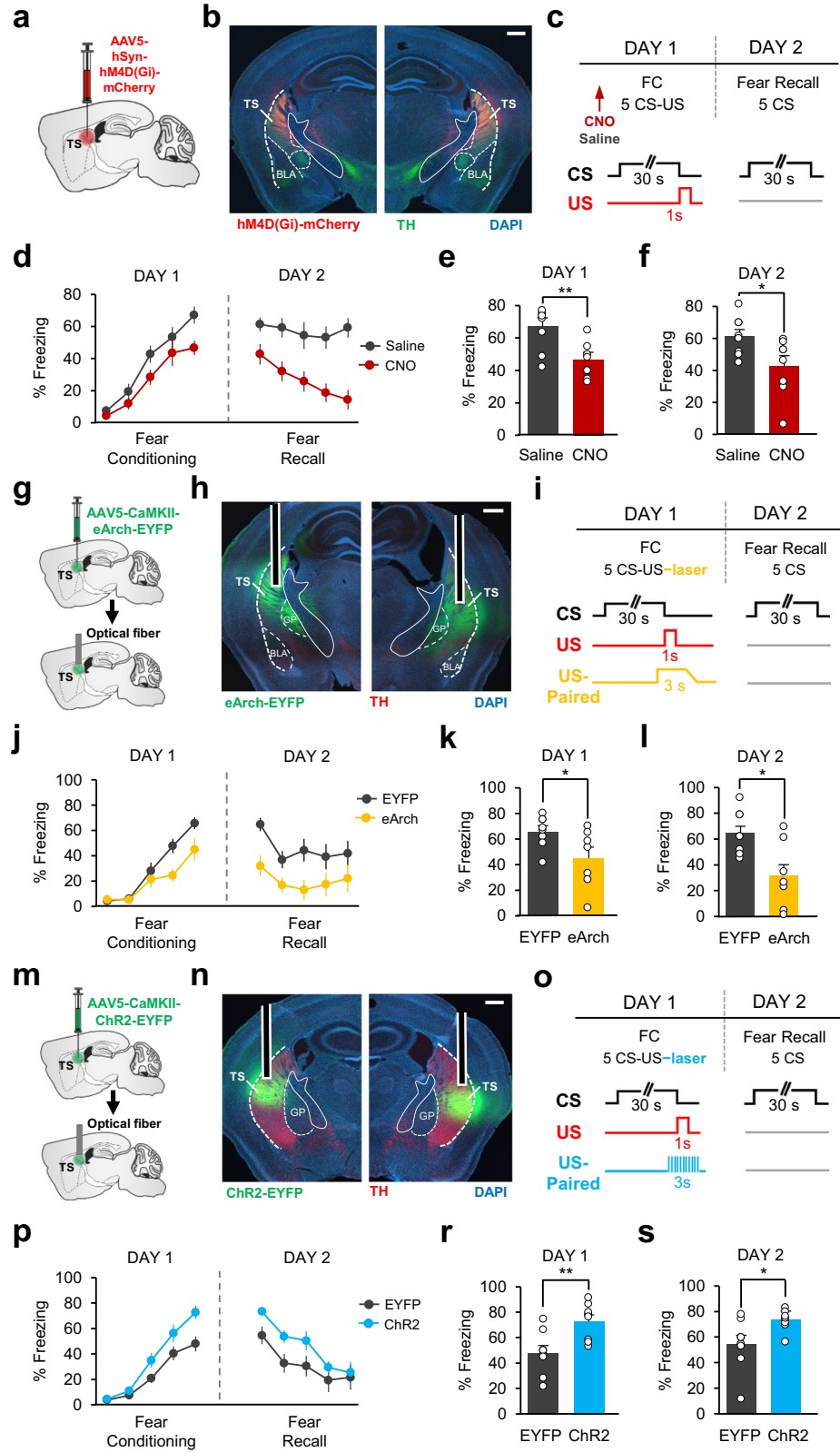

that TS-projecting DA neurons show weak reward responses and are not required for forming CS-reward associations. Furthermore, our results are also consistent with previous studies in head-fixed mice which showed that TS-projecting DA neurons had weak reward responses, did not exhibit RPE signaling during Pavlovian reward learning[35], and were not required for learning of reward value[16]. Consistently, we also found that increasing the reward value did not

affect the magnitude of reward responses in these DA neurons. Overall, these findings highlight the consistency of results across both head-fixed and freely-behaving conditions. Importantly, our results indicate an aversive bias in the responses of TS-projecting DA neurons. This distinguishes the responses of these DA neurons from other DA subpopulations that project to the striatum, for example, the ones projecting to the ventral subregion of NAc which have also

**Fig. 8 | Bidirectional manipulation of neuronal activity reveals the causal contribution of TS in associative fear learning. a** Schematic of the surgery. **b** Example histological image showing expression of hM4D(Gi)-mCherry (red), TH (green), and DAPI (blue) staining. Scale bar: 0.5 mm. **c** Top: schematic of the protocol. FC fear conditioning. Bottom: Schematic of CS and US presentations. **d** Freezing to the CS for saline ($n = 8$) and CNO ($n = 8$) groups (two-way repeated measures ANOVA, FC: main effect of group, $F_{1,56} = 5.08$, $P = 0.04$; Fear recall: main effect of group, $F_{1,56} = 19.61$, $P = 0.0006$). **e** Freezing to the last CS during FC (two-sided t-test, $P = 0.006$). **f** Freezing to the first CS during fear recall (two-sided t-test, $P = 0.03$). **g** Schematic of the surgery. **h** Example histological image showing expression of eArch-EYFP (green), TH (red), and DAPI (blue) staining. White vertical tracks indicate optical fiber placements. Scale bar: 0.5 mm. **i** Top: schematic of the protocol. Bottom: schematic of the US-Paired optogenetic inhibition. **j** Freezing to the CS for EYFP ($n = 8$) and eArch ($n = 7$) groups (two-way repeated measures ANOVA, group × trial interaction, $F_{4,52} = 4.40$, $P = 0.0039$). **k** Freezing to the last CS during FC (two-sided t-test, *$P = 0.04$). **l** Freezing to the first CS during fear recall (two-sided t-test, *$P = 0.01$). **m** Schematic of the surgery. **n** Example histological image showing expression of ChR2-EYFP (green), TH (red) and DAPI (blue) staining. White vertical tracks indicate the optical fiber placements. Scale bar: 0.5 mm. **o** Top: schematic of the protocol. Bottom: schematic of the US-Paired optogenetic excitation. **p** Freezing to the CS in EYFP ($n = 8$) and ChR2 ($n = 8$) groups (two-way repeated measures ANOVA, main effect of group, $F_{1,56} = 5.8$, $P = 0.03$). **r** Freezing to the last CS during FC (two-sided t-test, **$P = 0.008$). **s** Freezing to the first CS during fear recall (two-sided t-test, *$P = 0.03$). Error bars represent mean ± s.e.m. Source data are provided as a Source Data file.

been shown to be activated by foot shocks[28,68,69]. However, those DA neurons also exhibit strong, and even larger, responses to rewards compared to aversive stimuli[28,68], suggesting that they likely encode salience rather than aversion. Our results raise the question of whether activation of TS-projecting DA neurons induces aversion per se[18,33]. However, we found that exciting DA terminals in TS does not cause place avoidance or anxiety. Likewise, activation of these terminals at the end of the CS but in the absence of the US did not induce fear learning, suggesting that this DA input is likely not aversive by itself. Instead, our results suggest that DA input induces plasticity in TS that biases associative learning to cues that are specifically paired with aversive outcomes. Furthermore, we found that activation of DA input to TS during the CS enhances the CS-US association during fear learning, supporting the hypothesis proposed by Menegas et al.[35] that DA input to TS may enhance "CS associability".

Traditionally, the amygdala, in particular LA, is established to be the critical site for plasticity mediating the CS-US association during fear learning[5–8,10]. LA receives inputs from both the thalamus and the cortex, relaying information about the auditory CS[70]. Considerable evidence indicates that potentiation of CS-evoked responses in LA neurons underlies the acquisition of associative fear learning[71–76]. Moreover, plastic changes in the activity of neurons located in the central nucleus of the amygdala (CEA) have also been shown to underlie acquisition of fear memories[77–80]. In line with these findings, DA projections to the amygdala exhibit activation during fear learning[29,57,58]; and in particular, non-canonical dorsal tegmental DA neurons, that are located in the periaqueductal gray (PAG)/dorsal raphe (DR) and project specifically to CEA, have been shown to encode a PE signal to gate fear learning[29]. However, whether DA PE signaling outside the amygdala circuitry contributes to the acquisition of CS-US association during FC has remained largely unknown. Our findings reveal that DA neurons projecting to a brain structure outside the canonical amygdala circuitry encode a PE signal critical for driving associative fear learning.

While the role of the amygdala is well-established, it is becoming increasingly clear that acquisition of fear memories also involves plasticity in brain structures beyond the traditional amygdala circuitry[81–84]. Notably, despite earlier lesioning and anatomical studies hinting at the involvement of the posterior striatal areas, including the TS, in fear conditioning[85–87], the contribution of TS in fear learning remained largely elusive and has recently begun to be investigated[11,12]. TS, also known as the auditory striatum, is a distinct subregion of the dorsal striatum, characterized by the unique set of inputs it receives[13,14]. Similar to LA, TS receives auditory inputs from both the thalamus and the cortex[85,87]; and we here showed that auditory CS-evoked responses are potentiated also in the TS as the animals learned the CS-US association during FC, consistent with previous reports[11,12]. Previous studies have shown that activity of TS neurons is required for fear retrieval[11,12]. Importantly, we further demonstrate that neuronal activity in TS exhibits a PE-like activity pattern and is important for

acquisition of fear conditioning, indicating that a broader neural network than the amygdala circuitry is indeed involved in acquiring the CS-US association during fear learning.

One important question is whether and how TS is connected with the canonical amygdala fear learning circuitry. Although, TS receives strong input from the BLA[13], whether these amygdala projections to TS are critical for fear learning is largely unknown. Notably, BLA input to the posterior striatum have been shown to facilitate plasticity in corticostriatal synapses[88] suggesting that amygdala input might be critical for the plasticity underlying fear acquisition in TS neurons. Future studies investigating the role of amygdala input to TS during fear acquisition will be important to address this question. Conversely, although TS sends only sparse projections to the amygdala these projections have recently been shown to be important for fear retrieval[12], suggesting a direct connection to the canonical amygdala circuitry during fear conditioning. Further research on how TS input affects neuronal activity in distinct components of the amygdala circuitry will be crucial for elucidating the role of TS in fear retrieval. In addition, the role of TS projections to downstream structures outside the amygdala circuitry during fear acquisition remains elusive.

An interesting finding in our study is the PE-like activity pattern that we observed in TS. Similar to TS-projecting DA neurons, TS neuronal activity exhibited larger responses to the US at the beginning of FC when US presentation was unpredicted. As the CS-US association was learned, the US responses decreased, and conversely, responses to the CS gradually increased. Importantly, we found that PE coding during FC was dependent on the DA input to TS. Activity of TS-projecting SN neurons was shown to be important for the potentiation of CS responses during fear conditioning in a recent study. However, whether this was dependent on DA input to TS remained unclear as inhibition of neuronal activity in SN lacked DA neuron specificity[12]. Here, we demonstrated that ablation of specifically the TS-projecting DA neurons prevented potentiation of CS responses in parallel with impaired fear learning. In addition, we also found that the US responses did not decrease during the course of FC in mice with ablation of TS-projection DA neurons, consistent with the deficit in acquiring the CS-US association in these mice. Notably, responses to the first unpredicted US were not different between the lesioned and control mice indicating that the somatosensory inputs to TS relaying the US information were not affected by ablation of the DA input, but rather the predictive coding of US responses in TS was impaired. Together, these results reveal a PE-like activity in TS during fear learning and that these fear learning-induced activity patterns in TS were dependent on DA input during FC.

The principal neurons in TS are the medium spiny neurons (MSNs), categorized into two types based on their expression of dopamine D1 and D2 receptors, which constitute the direct and indirect pathways, respectively[14]. Whether and how D1- and D2-MSNs contribute to associative fear learning is an important question. A recent study investigating the role of these MSNs in the ventralmost portion of the TS, a region largely including the amygdalostriatal

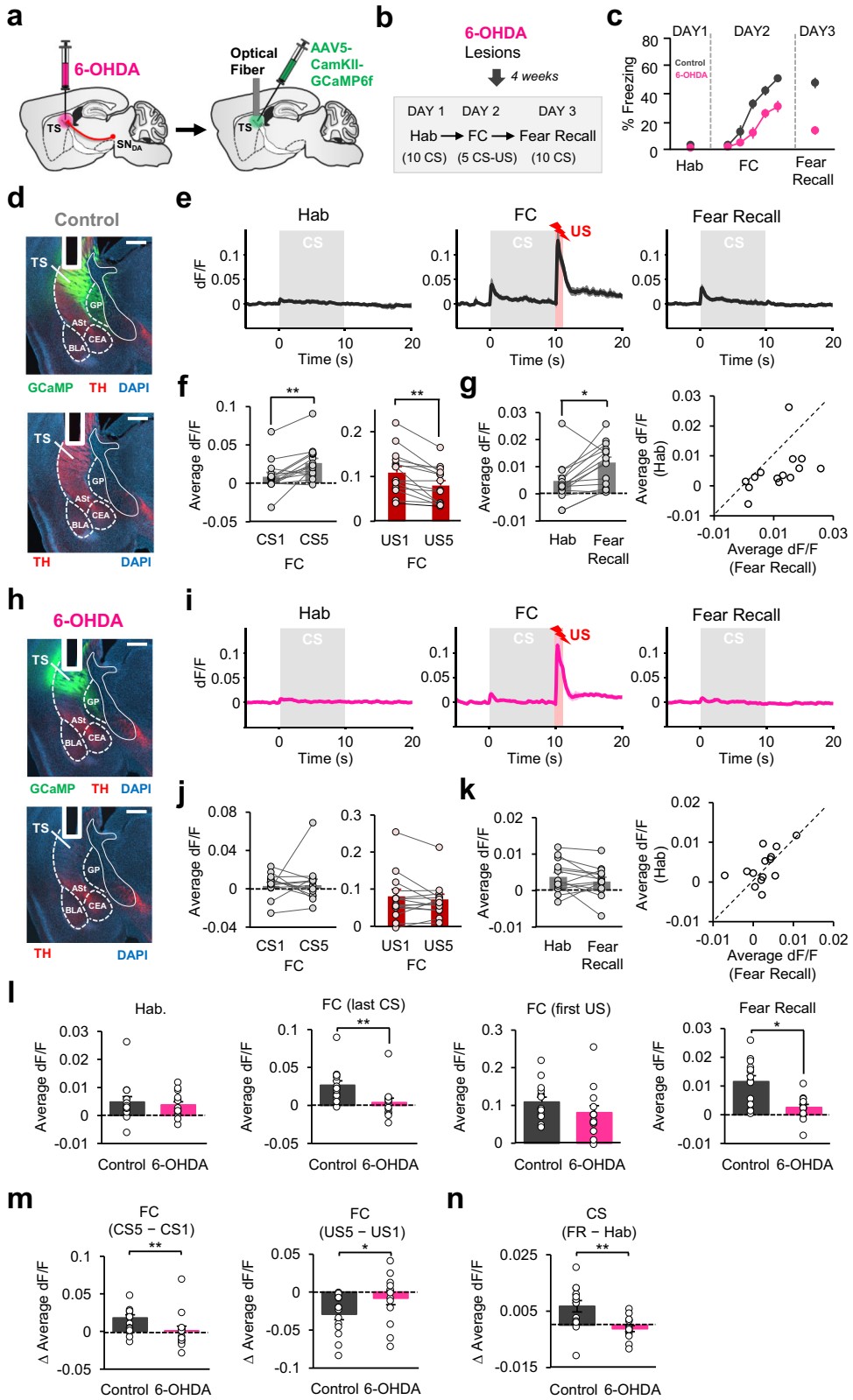

transition area (ASt), found that D1- but not D2-MSNs exhibited enhanced CS responses following fear conditioning[11]. Furthermore, while optogenetic inhibition of D1-MSNs impaired fear retrieval, inhibition of D2 neurons had no significant effect. However, the role of D1- and D2-MSNs in associative fear learning across the larger, more dorsal TS, where our study was focused, currently remains an open question. If D1-, but not D2-, MSNs are similarly involved in fear learning in the

dorsal TS, it is possible that the average population activity observed in our study was mainly driven by the activity of the D1-MSNs. It is also possible that our optogenetic and chemogenetic manipulations of all TS neurons actually influenced fear learning primarily through their effect on D1-MSNs. Future research investigating the activity of these neurons across the larger TS during fear learning will be necessary for addressing this question.

**Fig. 9 | DA input is critical for PE-like activity and potentiation of CS responses in TS during associative fear learning. a** Schematic of the surgeries. **b** Schematic of the procedure. Hab tone habituation, FC fear conditioning. **c** Freezing to the CS in control ($n = 10$) and 6-OHDA ($n = 8$) groups. Example image showing expression of GCaMP (green; top), TH (red), and DAPI (blue) staining of a control (**d**) and a 6-OHDA-lesioned (**h**) animal. White vertical track indicates the optical fiber placement. Scale bar: 0.5 mm. Average change in fluorescence around the CS presentation (gray area) in control (**e**, $n = 14$) and 6-OHDA (**i**, $n = 14$) groups. Left: average change in fluorescence during CS1 and CS5 of FC (**f**, control: **$P = 0.0011$, and **j** 6-OHDA: $P = 0.5$, two-sided signed-rank test). Right: average change in fluorescence during US1 and US5 of FC in the control (**$P = 0.0012$, two-sided signed-rank test) and 6-OHDA-lesioned ($P = 0.62$, two-sided signed-rank test) mice. Left: average change in fluorescence during the CS in the control (**g**, $n = 14$, *$P = 0.012$, two-sided signed-rank test) and 6-OHDA lesioned (**k**, $n = 14$) mice. Right: Scatter plots showing CS responses of each recording site in the control (**g**) and 6-OHDA (**k**) mice during Hab and Fear Recall. **l** Comparison of the average signal to the CS (left: Hab, $P = 0.76$; right: Fear recall, $P = 0.0054$, two-sided rank-sum tests), and CS (last-CS; $P = 0.0012$, two-sided rank-sum test) and unpredicted US (first-US, $P = 0.10$, two-sided rank-sum test) during FC (middle) between control and 6-OHDA groups. **m** Comparison of change in the average signal between first and last CSs (left) and USs (right) between control and 6-OHDA groups ($\Delta$CS: **$P = 0.004$; $\Delta$US: *$P = 0.04$, two-sided rank-sum test). **n** Comparison of change in the average signal between Hab and Fear Recall (FR) between the two groups (**$P = 0.0019$, two-sided rank-sum test). Shaded regions and error bars represent mean ± s.e.m. Source data are provided as a Source Data file.

PE signaling during FC has previously been demonstrated in LA neurons, where the PE coding was shown to set the associative memory strength[89]. In line with this notion, we here demonstrated that optogenetic excitation and inhibition of TS neuronal activity at the time of the US enhanced and impaired fear learning, respectively. These results indicate that the PE coding by TS neurons drives learning about the CS-US association. PE coding in LA neurons was shown to be mediated by a feedback neuronal circuitry involving projections from ventrolateral PAG (vlPAG) to LA[89,90], and likely also involves projections from the cerebellum to vlPAG[91,92]. Whether feedback neural circuits are recruited to drive PE coding in TS neurons, as well as in TS-projecting DA neurons, and the components of these neural circuits will be important questions for future research. Another important question that remains to be investigated is the differential contributions of LA versus TS neuronal activity in driving associative fear learning. Overall, our study reveals that a DA PE signal in a non-canonical nigrostriatal circuitry is crucial for driving associative fear learning.

## Methods

### Subjects

All procedures were conducted in accordance with the guidelines of the German Animal Welfare Act and were approved by the local authorities (Regierungspräsidium Darmstadt; protocol number 1256). Adult male C57BL/6N (Charles River or Janvier Labs) and heterozygous DAT-Cre mice[93] (backcrossed with C57BL/6N) aged older than 3 months at the start of experiments were used. All experimental groups were matched for age. For 6-OHDA lesioning, optogenetic and chemogenetic experiments, littermate mice were allocated to experimental and control groups. All mice were individually housed on a 12-h light/dark cycle. All experiments were performed during the light cycle.

### Viral constructs

We obtained AAV5-CAG-Flex-GCaMP6m-WPRE-SV40, pENN-AAV5-CaMKII-GCaMP6f-WPRE-SV40, AAV5-CAG-dLight1.1, AAV5-hSyn-hM4D(Gi)-mCherry and AAV5-hSyn-mCherry from Addgene, and AAV5-EF1a-DIO-hChR2(H134R)-EYFP, AAV5-EF1a-DIO-eArch3.0-EYFP, AAV5-EF1a-DIO-EYFP, AAV5-CaMKIIa-hChR2(H134R)-EYFP, AAV5-CaM-KII-ArchT-GFP, AAV5-CaMKIIa-EYFP and AAV5-CaMKII-GFP from the University of North Carolina Vector Core.

### Surgical procedures

Animals were anesthetized using isoflurane (1–2%) and placed in a stereotaxic frame. At the onset of anesthesia, all animals received intraperitoneal injections of atropine (0.05 mg/kg) and subcutaneous injections of carprofen (4 mg/kg) and dexamethasone (2 mg/kg). Eye gel was applied on the eyes to prevent dehydration of the cornea. Lidocaine cream was applied on the scalp as a local anesthetic. The animal's temperature was maintained for the duration of the surgical procedure using a heating blanket. Anesthesia levels were monitored throughout the surgery, and the concentration of isoflurane adjusted so that the breathing rate never fell below 1 Hz.

For GCaMP fiber photometry recordings of DA terminal activity in the TS, DAT-cre mice were injected with 0.5–1 µl of AAV5-CAG-Flex-GCaMP6m-WPRE-SV40 (final titer -1 × 10^13 pp per ml) in the SN (3.2 mm posterior to bregma, 1.2 mm lateral to the midline and 4.5 mm ventral to bregma)[94] at 50 nl/min using a 10 µl syringe with a 33-gauge needle controlled by an injection pump. The needle was left in place for an additional 10-15 min before slowly being withdrawn. Following infusion of the virus, optical fibers (400 µm core diameter, 0.48 NA, Doric Lenses) were slowly inserted through the craniotomy above the TS (AP: −1.3 mm, ML: 2.95 mm and DV: 2.75–3.0 mm)[94]. In a subset of animals, recordings were performed bilaterally. The optical fiber was then anchored to the skull using skull screws and dental cement (Paladur).

For GCaMP fiber photometry recordings of neuronal activity in the TS, wild-type C57BL/6 N mice were injected with 150 nl of AAV5-CaMKII-GCaMP6f-WPRE-SV40 (final titer -1 × 10^13 pp per ml) in the TS (AP: −1.3 mm, ML: 2.95 mm and DV: 3.25 mm) at 50 nl/min using a 10 µl syringe with a 33-gauge needle controlled by an injection pump. The needle was left in place for an additional 10–15 min before slowly being withdrawn. Following infusion of the virus, optical fibers (400 µm core diameter, 0.48 NA, Doric Lenses) were slowly inserted through the same craniotomy to a depth of 2.75–3.0 mm below the bregma. The optical fiber was then anchored to the skull using skull screws and dental cement (Paladur).

For dLight fiber photometry experiments, wild-type C57BL/6N mice were injected with 150–300 nl of AAV5-CAG-dLight1.1 (final titer 4.2 × 10^12 pp per ml) in the TS (AP: −1.3 mm, ML: 2.95 mm and DV: 3.25 mm) as described above. Following infusion of the virus, an optical fiber (400 µm core diameter, 0.48 NA, Doric Lenses) was slowly inserted through the same craniotomy to a depth of 2.75–3.0 mm below the bregma. In a subset of animals, recordings were performed bilaterally. The optical fibers were then anchored to the skull using skull screws and dental cement (Paladur).

6-OHDA lesioning of TS-projecting DA neurons was performed as previously described[16,95]. Thirty min before the surgery, wild-type C57BL/6 N mice received intraperitoneal injections of a solution (10 ml/kg) containing desipramine hydrochloride (2.5 mg/ml, Sigma-Aldrich) and pargyline hydrochloride (0.5 mg/ml, Sigma-Aldrich) dissolved in 0.9% saline (pH 7.4), in order to prevent uptake of 6-OHDA in noradrenaline neurons. 6-OHDA (10 mg/ml; 6-OHDA hydrobromide, Sigma-Aldrich) was dissolved in a vehicle solution immediately before the surgeries. The vehicle contained saline (0.9%) and a small amount of ascorbic acid (0.2%) to prevent oxidation of 6-OHDA. To further minimize oxidation of 6-OHDA, the solution was kept on ice, wrapped in aluminum foil and used within 4 h after preparation. During the surgery, animals were injected bilaterally with 150–300 nl of the 6-OHDA solution or the vehicle in the TS using the coordinates described above. The injections were performed at 50 nl/min speed using a 10 µl syringe with a 33-gauge needle, which was controlled by an injection pump. The needle was left in place for an additional 10-

15 min before slowly being withdrawn. The scalp incision was then closed using sutures. In experiments with GCaMP recordings, mice received virus injections and optical fiber implantation as described above one week after 6-OHDA injections. In a subset of animals in the control and 6-OHDA groups, recordings were performed bilaterally.

For optogenetic excitation or inhibition of DA terminals in the TS, DAT-cre mice were injected bilaterally in the SN with 0.5–1 µl of AAV5-EF1a-DIO-hChR2(H134R)-EYFP (final titer 4.4 x $10^{12}$ pp per ml), AAV5-EF1a-DIO-eArch3.0-EYFP (final titer 5.6 × $10^{12}$ pp per ml) or AAV5-EF1a-DIO-EYFP (final titer 4.3 x $10^{12}$ pp per ml) per hemisphere using the coordinates described above. Virus injection was performed as described above and followed by implantation of optical fibers (200 µm core diameter, 0.22 NA, Thorlabs) bilaterally above the TS (AP: −1.3 mm, ML: 2.95 mm) to a depth of 2.75–3.0 mm below bregma. The optical fibers were then anchored to the skull using skull screws and dental cement (Paladur).

For optogenetic excitation or inhibition of neuronal activity in the TS, wild-type C57BL/6 N mice were injected bilaterally in the TS with 150 nl of AAV5-CaMKIIa-hChR2(H134R)-EYFP (final titer 4.1 x $10^{12}$ pp per ml), AAV5-CaMKII-ArchT-GFP (final titer 5.2 ×$10^{12}$ pp per ml) or AAV5-CaMKIIa-EYFP (final titer 3.6 x $10^{12}$ pp per ml) per hemisphere using the coordinates described above. Virus injection was performed as described above. Optical fibers (200 µm core diameter, 0.22 NA, Thorlabs) were then implanted bilaterally above the TS using the coordinates described above. The optical fibers were then anchored to the skull using skull screws and dental cement (Paladur).

For chemogenetic experiments, C57BL/6N mice were bilaterally injected with 150 nl of AAV5-hSyn-hM4D(Gi)-mCherry (final titer 8.6 x $10^{12}$ particles per ml) or AAV5-hSyn-mCherry (final titer 2.3 × $10^{13}$ particles per ml) per each hemisphere in the TS using the coordinates described above. Virus injection was performed as described above. The scalp incision was then closed using sutures at the end of the surgery.

### Behavior

**Fear conditioning.** Fear conditioning and fear recall took place in two different contexts (A and B). Context A consisted of a square chamber with an electrical grid floor (Med Associates) used to deliver the footshock US. Context B consisted of a white teflon cylindrical chamber with bedding material on the floor. The chambers were located inside a sound attenuating box and were cleaned with 1% acetic acid before and after each session. Before fear conditioning experiments started, all mice were habituated to handling and being connected to the patch cord and habituated to contexts A and B for 10-15 min each in a counterbalanced fashion. On day 1, mice received a tone habituation session which started following a 2 min baseline period in context A and consisted of 10 presentations of the CS (4 kHz tone, 75 dB) with a random intertrial interval (ITI) of 40–120 s. On day 2, mice underwent fear conditioning consisting of five pairings of the CS with a US (1 s footshock, ITI: 40-120 s). The CS was 10 s and 30 s long in photometry and optogenetic/chemogenetic experiments, respectively. In 6-OHDA experiments, both 10 s and 30 s long CSs were used. Comparable results were obtained with both CS durations ($p > 0.05$) and hence the data was pooled. Furthermore, in 6-OHDA experiments, a group of mice underwent visual fear conditioning in which the CS was a 30 s long light cue (yellow LED diode, 0.25 W). The US intensity was 0.4–0.5 mA for photometry and 0.5 mA for 6-OHDA, optogenetic inhibition and chemogenetic experiments. For optogenetic excitation experiments where mice received a weak training, the US intensity was 0.35 mA. The onset of the US coincided with the offset of the CS. On day 3, mice received a fear recall session consisting of 10 presentations of the CS alone in context B for photometry experiments. For 6-OHDA, optogenetic and chemogenetic experiments, fear recall test consisted of 5 presentations of the CS on day 3. For contextual fear conditioning, mice received context habituation (context A) for 10 min on day 1. On day 2, following a 2 min baseline period mice received 5 presentations of the US (0.5 mA) with a random intertrial interval (ITI) of 40–120 s in context A. On day 3, mice were placed back to context A for 10 min for their context testing. For the unpaired training protocol, mice received five presentations of the CS and the US in an explicitly unpaired fashion (ITI: 40-120 s) on day 2. At the end of fiber photometry experiments, one subset of mice was further tested for predicted versus unpredicted US presentations (5 CS-US pairings and 5 USs presented in a random order). Another subset of mice were further trained on a partial fear conditioning protocol in which they received 5 CS-US pairings and 5 CS presentations in a random order to examine US omission responses. Throughout the experiments, the behavior of mice was recorded to video and scored by experienced observers blind to the experimental condition. Behavioral freezing, defined as the absence of all bodily movements except breathing-related movement[96], was used as the measure of fear.

**Reward Tasks.** Mice in the photometry experiments underwent a reward task following the fear conditioning protocol. To motivate animals, their access to water was restricted until they reached 85% of their body weight. Animals were placed in an operant chamber containing a liquid delivery port. Nosepokes into the delivery port that followed the previous nosepoke by at least a variable inter-trial interval of 3–5 s triggered the delivery of liquid reward (10% or 25% sucrose solution). Animals were first trained on 90% probability of reward delivery on day 1, in which they quickly learned the task. The next day, photometry recordings were performed while rewards were delivered randomly at a 50% probability, making reward delivery unpredictable. To test whether prior fear conditioning affected reward responses, a separate cohort of animals underwent the reward task both before and after fear conditioning. Reward responses before fear conditioning were comparable to ones after fear conditioning ($n = 7$; $p = 0.37$, signed-rank test).

In 6-OHDA experiment, mice underwent a reward learning task for 5 consecutive days. Animals were placed in an operant chamber containing a liquid delivery port. Each session consisted of 50 pairings of a CS (8 kHz tone, 75 dB, 5 s long) and a reward (10% sucrose solution) with a random ITI of 20–30 s. If the animal accessed the reward port during the CS presentation, a 10 µl reward was delivered.

**Real-time place preference test.** At the end of fear conditioning protocol, a subset of mice in the optogenetic experiments underwent place preference, open field, and elevated plus maze tests. Real-time place preference test was conducted in a custom-made chamber (50 × 50 x 50 cm, wooden gray box) divided into two compartments. The test consisted of two 10 min phases (Fig. 5m). During the first phase, one side of the chamber was randomly assigned as the laser ON side. Mice were individually connected to the patch cords and placed in the laser OFF side of the chamber. Each time the mouse entered the laser ON side laser light was delivered until the mouse crossed back to the OFF side. In the second phase, the sides were switched and the previously laser OFF side became laser ON side in order to counterbalance each side.

**Open field test.** The custom made open field chamber (50 × 50 x 50 cm, wooden gray box) was divided into a central area (center 25 × 25 cm) and an outer area (periphery). The open field test consisted of a 9 minute session with three alternating 3-minute epochs (OFF-ON-OFF epochs) in which laser was delivered during the ON epoch (Fig. 5n).

**Elevated Plus Maze (EPM).** EPM consisted of two open arms (30 × 5 cm), two closed arms (30 × 5 × 15 cm) and a central area (5 × 5 × 5 cm). The maze was placed 40 cm above the floor. Mice were individually connected to the patch cords and placed in the center. The test consisted of a 9 minute session with three alternating 3-minute epochs

(OFF-ON-OFF epochs) in which laser was delivered during the ON epoch (Fig. 5o).

## GCaMP and dLight recordings using fiber photometry

Animals were injected with viral vectors and implanted with optical fibers in the TS, as described above. After a waiting period of 3-4 weeks to allow for surgical recovery and virus expression, mice were connected to 400 µm patch cords (Doric Lenses). Fluorescence was measured by delivering 465 nm excitation light through the patch cord and separating the emission light at 525 nm with a beamsplitter (Fluorescence MiniCube FMC3, Doric Lenses). The emission light was collected using a Femtowatt Silicon Photoreceiver (Model # 2151, Newport). The voltage output of the photoreceiver was then digitized at 2 kHz (Digital Lynx SX, Neuralynx). After animals were habituated to handling and being connected to the patch cord, they underwent the behavioral protocol as described above. Photometry recordings were performed throughout the protocol.

## Analysis of fiber photometry data

The voltage output of the photoreceiver, representing fluctuations in fluorescence, was downsampled to 10 Hz. The change in fluorescence evoked by the CS (dF/F) was then calculated by subtracting from each trace the baseline fluorescence (average during the 5 s before CS onset) and dividing it by the baseline fluorescence. dF/F traces were then averaged separately for each animal for tone habituation (Hab: average of 10 CSs), fear conditioning (Fear Cond., average of 5 CSs) and fear recall (average of 10 CSs). To examine responses to the CS, we further averaged dF/F values in the 5 s following CS onset in each session. To quantify responses to the US, we averaged the dF/F values at the time of the US (1 s) during Fear Cond. for each animal. To examine the change in responses to the CS and the US during Fear Cond., we compared responses to the first and last CSs and USs for each animal. To examine responses to reward, average dF/F was calculated for rewarded trials using the baseline fluorescence 3 s before noseport entry. Reward responses were quantified by averaging the dF/F in the 3 s following noise port entry for each animal.

## Chemogenetic experiments

Three to four weeks after viral injections, mice underwent the chemogenetic experiment. Thirty min before the fear conditioning session, mice expressing the inhibitory DREADD hM4D(Gi)-mCherry or the control fluorophore mCherry received systemic injections of the DREADD agonist clozapine N-oxide (CNO, Sigma-Aldrich; 1–1.5 mg/kg dissolved in saline) or saline. Fear conditioning was performed as described above. The next day, animals underwent the fear recall test drug free.

## Optogenetic experiments

For bilateral optogenetic manipulations during behavior, the implanted optical fibers (200 µm core diameter, 0.22 NA, Thorlabs) were connected to 200 µm patch cords (Thorlabs) with zirconia sleeves and the patch cords were connected to a light splitting rotary joint (FRJ 1x2i, Doric Lenses) that was connected to a laser with a 200 µm patch cord (Thorlabs). For mice expressing the light-activated excitatory opsin ChR2 (ChR2-EYFP) and their EYFP controls, blue light pulses were delivered from a 473 nm laser (LuxX473, Omicron). Laser power at the tip of the optic fiber was 5–10 mW. DA terminals in TS were excited at the time of the CS (CS-Paired), the US (US-Paired), during the ITIs (ITI Control) or in the absence of the US (No-US Control), and TS neuronal activity was excited at the time of the US (US-Paired) during fear conditioning. For CS-Paired animals, 5-ms light pulses were delivered at 20 Hz during the CS presentation. For US-Paired animals, 5-ms light pulses were delivered at 20 Hz for 1 or 3 seconds. For ITI Control group, 5-ms light pulses were delivered at 20 Hz for 30 s during the ITIs. For No-US Control group, 5-ms light pulses were delivered at 20 Hz for 3 s at the end of the CS. The laser was turned on 1 s before the US onset to 1 s after the US offset.

For mice expressing the light-activated inhibitory opsin eArch (eArch-EYFP) or ArchT (ArchT-GFP) and their EYFP or GFP controls, yellow light pulses were delivered from a DPSS 594 nm laser (Omicron). Laser power at the tip of the optic fiber was 5–10 mW. DA terminals in TS or TS neurons were inhibited at the time of the US from 1 s before to 2 s after CS offset. The laser was then turned off gradually using a 2 s ramp to avoid rebound excitation.

## Histology

At the end of the experiments, mice were deeply anesthetized with sodium pentobarbital and were transcardially perfused with 4% paraformaldehyde and 15% picric acid in phosphate-buffered saline (PBS). Brains were removed, post-fixed overnight and coronal brain slices (60 µm) were sectioned using a vibratome (VT1000S, Leica). Standard immunohistochemical procedures were performed on free-floating brain slices. Briefly, sections were rinsed with PBS and then incubated in a blocking solution (10% horse serum, 0.5% Triton X-100 and 0.2% BSA in PBS) for 1 h at room temperature. Slices were then incubated in a carrier solution (1% horse serum, 0.5% Triton X-100 and 0.2% BSA in PBS) containing the primary antibody overnight at room temperature. The next day, the sections were washed in PBS and then incubated in the same carrier solution containing the secondary antibody overnight at room temperature. The following primary antibodies were used: polyclonal rabbit anti-tyrosine hydroxylase (TH, catalog # 657012, 1:1000, Calbiochem), monoclonal mouse anti-TH (catalog # MAB318, 1:1000, Millipore), polyclonal guinea pig anti-TH (catalog # 213004, 1:1000, Synaptic Systems), polyclonal rabbit anti-GFP (catalog # A11122, 1:1000, Life Technology), polyclonal chicken anti-GFP (catalog # AB13970, Abcam), and mCherry monoclonal anti-rat (catalog # M11217, 1:1000) Invitrogen). The following secondary antibodies were used: Alexa Fluor 568 goat anti-rabbit (catalog # A11011, 1:1000, Thermo Fisher Scientific, Invitrogen), Alexa Fluor 568 goat anti-mouse (catalog # A11004, 1:1000, Thermo Fisher Scientific, Invitrogen), Alexa Fluor 568 goat anti-guinea pig (catalog # A11075, 1:1000, Invitrogen), Alexa Fluor 405 goat anti-rabbit (catalog # A31556, 1:750, Invitrogen), Alexa Fluor 488 goat anti-rabbit (catalog # A11008, 1:1000, Thermo Fisher Scientific, Invitrogen), Alexa Fluor 488 goat anti-chicken (catalog # AB150173, 1:1000, Abcam), Alexa Fluor 568 goat anti-rat (catalog # A11077, 1:1000, Invitrogen). For DAPI staining, sections were incubated for 10 min in 0.1 M PBS containing 0.02% DAPI (catalog # D1306, Molecular Probes, Invitrogen). Finally, all sections were washed with PBS, mounted on slides embedded with a mounting medium for fluorescence (VECTASHIELD®, Vector Laboratories), and coverslipped.

## Statistics

Data were statistically analyzed using GraphPad Prism (GraphPad Software) and MATLAB (Mathworks). All statistical tests were two-tailed and had an α level of 0.05. All error bars show s.e.m. All ANOVAs were followed by Bonferroni *post hoc* tests if significant main or interaction effects were detected. No statistical methods were used to predetermine sample size, but our sample sizes were similar to those generally used in the fear conditioning field. Animals were randomly assigned to experimental groups before the start of each experiment after ensuring that all experimental groups were matched for age. For 6-OHDA lesioning, chemogenetic and optogenetic experiments, experimental and control groups were matched from littermate mice. All results were obtained using groups of mice that were run in several cohorts.

## Reporting summary

Further information on research design is available in the Nature Portfolio Reporting Summary linked to this article.

## Data availability

All data in the figures and supplementary data are provided in the Source Data file. Raw data are available from the corresponding author upon request. Source data are provided with this paper.

## Code availability

This paper does not report original code. The custom codes used for data analysis in this study are available from the corresponding author upon request.

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

## Acknowledgements

We would like to thank Jochen Roeper for his support; Beatrice Fischer, Jasmine Sonntag, Sebastian Betz, Günther Amrhein and Thomas Wulf for technical assistance; and Torfi Sigurdsson for helpful discussions. This work was supported by the Deutsche Forschungsgemeinschaft (DFG Grant DU 1433/5-1 to S.D.).

## Author contributions

D.Z., X.I.S-H., and S.D. designed the experiments. D.Z., X.I.S-H., E.S.D.B., L.R., and S.D. performed the experiments. D.Z., X.I.S-H., and S.D. analyzed the data. X.I.S-H. helped with the supervision of E.S.D.B., L.R., and D.Z. D.Z., X.I.S-H., and S.D. wrote the paper with input from all authors. S.D. supervised the study and obtained funding.

## Funding

## Competing interests

The authors declare no competing interests.
