## [Transparent Peer Review file · Nature Communications]

Dopamine Prediction Error Signaling in a Unique Nigrostriatal Circuit is Critical for Associative Fear Learning

Corresponding Author: Dr Sevil Duvarci

Version 0:

Reviewer comments:

Reviewer #1

(Remarks to the Author)

In this study, Zafiri and colleagues examined the role of dopamine neurons projecting to the posterior tail of the striatum (TS) in Pavlovian fear conditioning. The authors examined the activity patterns of TS-projecting dopamine neurons as well as dopamine release in TS, and showed that dopamine inputs to the TS encode prediction error signals for fear learning (threat prediction error or aversive prediction error). Furthermore, suppression and enhancement of dopamine signals in TS reduced and facilitated fear learning. The authors also show that the population activity of TS neurons exhibited prediction error-like activity, which depended on intact dopamine inputs to the TS. Together this study demonstrates that prediction error-like dopamine signals for threat in the TS play a critical role in Pavlovian fear conditioning.

Pavlovian fear conditioning has been studied extensively, yet previous studies focused mainly on the amygdala. This study, together with other recent studies (Kintscher et al., 2023; Chen et al., 2023), identifies the TS as a novel brain area involved in fear conditioning. It is important that the authors show that plastic change in neuronal activity occurs in a dopamine-dependent manner and identify a critical neuromodulatory input that appears to convey prediction error-like signals for fear learning. The results complement recent studies on TS, highlighting its role in avoidance learning. This study overall presents important results, and the results are presented clearly. The conclusions are supported by proper control experiments (e.g. unpaired control, reward learning etc.), yet some experiments lack important controls. These control experiments should be performed to make the results of these experiments interpretable.

1. In fiber photometry experiments, movements of the animal cause artifactual signals. To control for this, the authors need to perform some control experiments such as monitoring signals from control animals expressing EGFP/mCherry or monitoring at the isosbestic wavelength.

2. In experiments using DREADD (pharmacogenetics), the injection of clozapine-N-oxide (CNO) itself might have effects on behavior. The authors present only control experiments using saline injections, and this does not address this potential confound. The authors need to perform control experiments injecting CNO into animals injected with a control virus.

3. Page 10. Line 14. "we hypothesized that DA projections to TS might be important for associating specifically the sensory cues with aversive outcomes. To address this, we next investigated whether TS-projecting DA neurons were necessary for learning the association between the context and the aversive US". A context is ultimately determined by sensory inputs. The dichotomy between sensory cues versus context, therefore, appears to be ill-defined. For example, the presence of long-lasting olfactory cues might have rescued from the effect of 6-OHDA. The conclusions that can be made from this experiment, therefore, need to be discussed more carefully.

4. Page 18, Line 18. The area identified by de Jong et al. (2019) is the ventromedial part of NAc. This study also identified the lateral NAc as encoding canonical reward-related signals. It seems that the other two studies cited here (Yuan et al., 2019; Salinas-Hernandez et al., 2022) found slightly different, more complex distribution of responses. Is there a consensus or is the description used here ("lateral and ventral") a good summary of those dopamine inputs activated by both rewarding and aversive stimuli?

Reviewer #2

(Remarks to the Author)

The authors show that activity in a population of dopaminergic neurons projecting to the tail of the striatum is required for cued, but not contextual fear learning. The authors also show that this circuit is not required for cued-reward learning or for retrieval of cued fear. This work also elegantly shows that the tail of the striatum shows aversion prediction error that requires this dopaminergic innervation. The calcium imaging data demonstrating the mechanisms behind the prediction error signal are well-done and informative. These data are timely and are of interest to the readership of the journal. However, there are important issues that must be addressed, as explained below.

Major Points

1. Please write the correlation r and p values for all scatterplots in all figures, such as Figure 2E, for example.
2. Figure 5 shows that optogenetic activation of dopaminergic terminals in TS paired with US or CS increase fear learning. Please add a control showing the effect of activating these terminals during five 30-second epochs outside of CS or US, to demonstrate if the effect shown in 5G and 5H is temporally specific.
3. The authors state that temporally precise signaling of the TS is important for fear learning. It is thus puzzling why Figure 7A-F is done with chemogenetics, which is a method that has very low temporal specificity. Please replicate these results using inhibitory optogenetics instead.
4. Please show if optical inhibition of dopaminergic fibers in TS paired with either US or CS impairs fear learning.
5. Please add text to the discussion explaining how and if the circuit studied here is connected to the canonical amygdala fear learning circuitry. Do the authors propose that this circuit affects the amygdala or that it acts independently of the amygdala to modulate fear learning?
6. Please show if the major perturbational experiments done also affect overall speed and anxiety. Anxiety (but not speed) data was shown in Figure 5. The potential effects of other manipulations on anxiety and speed were not shown for other experiments controlling neural activity, such as those in Figure 7. As the authors are aware, changes in overall anxiety and speed can be major confounds when the main metric plotted is freezing.

Reviewer #3

(Remarks to the Author)

In this study by Zafiri et al., the authors investigate the role of SNc dopamine projections to the tail of the striatum for regulating associative fear learning. In general, the studies are well designed, well controlled, and well executed and provide novel information on the role of dopamine in this specific context in associative fear learning. Although my enthusiasm for the results of the study is generally high, I have some major concerns relating to the interpretation of some results and the overall contextual framing of the manuscript.

Specific comments:

1. My biggest concern and disappointment related to this study is the following statement: "although these DA neurons have been implicated in aversion their role in driving fear learning has remained elusive" (lines 19-20). This statement is false. Either the authors are unfamiliar with the literature or have attempted to build a false narrative that borders on the unethical. Aside from the few studies referenced by the authors that relate mostly to fear extinction, numerous lesion studies, pharmacological studies, and electrophysiological studies that have implicated the midbrain dopamine system in fear learning (see Lamont and Kokkinidis, 1998; Nader and LeDoux, 1999; Guarraci and Kapp, 1999; Greba and Kokkinidis 2000, Greba et al., 2001 just to name a few from 100s) there have also been numerous studies directly implicating midbrain dopamine neurons in associative fear learning and the encoding of fear related information during associative learning. Fadok et al., 2009 demonstrated that dopamine production is essential for associative fear learning, and further showed that dopamine production in projection to the amygdala and striatum are important (Fadok et al., 2010 and 2011). Electrophysiological responses and calcium dynamics in response to aversive and fear-inducing USs and the acquisition of CS responding across multiple paradigms have also been reported (Chiodo et al. 1979; Schultz and Romo 1987; Mantz et al. 1989; Ungless et al. 2004; Brischoux et al. 2009; Mirenowicz and Schultz 1996; Guarraci and Kapp 1999; Joshua et al. 2008; Matsumoto and Hikosaka 2009; Wang and Tsien 2011; Zweifel et al. 2011; Cohen et al. 2012; Gore et al., 2014; Luo et al., 2018; Salinas-Hernandez et al., 2018; Jo et al., 2018; Steinberg et al., 2020; and Cai et al., 2020), as have studies directly measuring transient dopamine release during fear-related learning (Roitman et al. 2008 and Badrinarayan et al. 2012). At the functional level, temporally precise inhibition of VTA dopamine neurons at the onset of a CS prevents discriminatory fear learning, as does inhibition of dopamine terminals in the CeA (Jo et al., 2018). Inhibition of SNc neurons projecting the CeA also significantly attenuates fear learning (Steinberg et al., 2020). The authors need to frame their study in the context of these prior studies on the role of dopamine neurons and associative fear learning, not just their own work and literature reviews. In my opinion, the findings presented here make several novel contributions to the field relating to the role of the tail of the striatum specifically; however, it does not benefit them, and is a major disservice to the field and those that have worked hard to advance this field, to claim to have resolved a previously "elusive" function of dopamine in fear learning. They are not the first to implicate dopamine in very specific aspects of associative fear learning, nor are they the first to provide a detailed analysis of the role of dopamine in the encoding of a conditioned fear stimulus and the

- neuroanatomical function of dopamine circuitry in learning the relationship between a CS and a fear US, give me a break.
2. The authors repeatedly refer to the observed responses that diminish to the US and increase to the CS as prediction error like. This is only half the equation the authors need to do random omissions to assess prediction errors.
 3. It is unclear based on the description of the design of the 50% reward probability how this is a measure of uncertainty that would be predicted to cause a change in dopamine release. Where the mice conditioned for numerous days at 90% probability then switched to 50% probability with a randomized design. If they had one day of 90% training then switched to 50% probability with every other nose poke resulting in reward delivery then this is just FR2, not an assessment of uncertainty. It is also not ideal to perform appetitive conditioning on mice after fear conditioning as this diminishes their overall hedonic state.
 4. Assessment of motivational salience requires varying the magnitude of the unconditioned stimulus and assessing the relationship between the observed signals, the conditioned behavioral responses, and the magnitude of the US.
 5. The tail of the striatum contains D1 and D2 receptor expressing neurons. Presumably these neurons respond differently to dopamine (stimulation versus inhibition); thus, it is unclear how bulk stimulation or inhibition of all TS neurons influences associative learning. This is not a major criticism, but the caveat should be addressed.

Reviewer #4

(Remarks to the Author)

In the present paper, Zafiri et al. investigated the role of dopamine (DA) neurons projecting to the tail of the striatum (TS) in associative fear learning. First, they used fiber photometry to record from three different indicators – GCaMP in TS-projecting DA neurons, GCaMP in TS neurons, and dopamine sensor in the TS – during associative fear learning (a tone paired with a footshock) and a following fear recall session. They found remarkably similar activity in each case: all showed characteristics of an aversive prediction error (PE) signal. Next, they established that activity of TS-projecting DA neurons and TS neurons are necessary for associative fear learning. Lesioning TS-projecting DA neurons before fear conditioning or chemogenetically inhibiting TS neurons during fear conditioning led to impaired learning during conditioning and recall. Furthermore, optogenetic excitation of TS-projecting DA neurons during the CS or the US, or excitation of TS neurons during the US, strengthened the effect of fear conditioning, enhancing freezing behavior. Furthermore, when TS-projecting DA neurons were lesioned, prediction-error like signaling in TS neurons was abolished, suggesting that the emergence of PE-like activity patterns in TS relies on dopaminergic input. Interestingly, despite its key role in associative fear learning, TS DA was not necessary for contextual fear learning, as animals with TS DA lesions still showed robust freezing behavior when placed back into the fear conditioning context.

Overall these experiments were logically constructed and comprehensive for addressing the authors' posed questions, including appropriate controls in most parts. The results were also clearly presented. This study's findings overlap considerably with past works (Menegas et al., 2017, 2018), which showed prediction error-like coding in TS DA neurons during classical conditioning and TS DA's role in learning of aversive airpuff, and this diminishes the novelty of the work. However, considering controversies in roles of TS, the beautiful set of results in the present study will contribute to the field. Further, the authors present several useful advances, including clarifying the role of TS DA in associative and contextual fear learning and the specific role in learning but not expression of conditioned fear. The work would be considerably strengthened and rendered more novel with the addition of more detailed analyses of the behavioral paradigms, a few more targeted experiments, and discussion of the novel findings, suggestions for which are contained below along with general issues.

Major concerns

1. The lack of an impairment in the contextual fear learning paradigm following TS DA lesion is interesting. Contextual fear learning depends to some extent upon associative fear learning: animals must at least associate sensory stimuli in the environment with the unconditioned stimulus to develop this fear. One possibility is that TS DA is required for learning to associate auditory stimuli with aversive outcomes, but not stimuli of other sensory modalities. To dissociate auditory vs other modalities, and cued vs contextual learning, authors should perform their fear conditioning experiments with another sensory modality as the CS – for instance, a flash of light.

2. The separation between fear conditioning and reward conditioning is interesting. Because it is very important, authors need more careful examination using comparative conditions. For comparison of neural responses, while the "unpredicted shock" is almost unpredicted, the "unpredicted reward" in Figure 2I is 50% predicted. Therefore, the response to 50% predicted reward in this experiment is likely to be not the maximum reward response of dopamine in TS. Moreover, the dopamine response at the reward timing in this task is difficult to interpret because it includes both the reward response and the nose entry response, which could reflect an increase in reward expectation. Because authors chose to use operant task for reward while footshock is passive, it is difficult to directly compare. Again, similar to the Concern 1, to dissociate classical vs instrumental learning, and reward vs threat learning, I recommend that the authors reconsider the task designs to make them comparable.

For behavioral effects of dopamine manipulation, it is also important to use similar tasks rather than use passive fear conditioning and operant reward acquisition. Further, it is also important to analyze behaviors more systematically, instead of relying on only one measurement (whether reward is acquired after cue). For example, the lesioned animals could get the same number of rewards following tone without forming a clear association – they could simply continuously nose-poke into the port, whether or not the tone was playing. The authors should show other behavioral correlates such as number of pokes into the port following the tone and in the inter-tone interval, length of time in the port, and velocity, to investigate such possibilities.

3. Authors should frame their study in a more accurate way in the fields since many findings are overlapped with findings in

previous studies by Menegas et al., 2017, 2018, Kintshcher et al., 2021, and Chen et al., 2023. For example, Menegas et al. found that TS-projecting dopamine neurons play a critical role in learning of aversive air puff, using ablation with 6OHDA. They also found that ablation of TS-projecting dopamine neurons does not affect learning of reward value in exactly the same task as threat learning. The present study did not discuss these although they confirmed these previous findings.

For the neural activity, Menegas et al. found that these dopamine neurons signal threat or physical salience of external stimuli, but not aversiveness per se because the response was selective with salient external stimuli but not with aversive bitter taste. These studies also found that dopamine responses have some characteristics of prediction error, but not perfect prediction error because they did not show inhibitory responses to omission. The present study only examined an electric shock and did not examine responses to other aversive stimuli or omission of aversive stimuli. While results in the present study is consistent with the previous proposal of threat prediction error, it is not justified that these dopamine neurons signal aversiveness or aversive prediction error in contrast to previous studies.

Similarly, although Kintshcher et al. and Chen et al. examined TS roles in fear conditioning, there is very limited discussion of these studies. Especially, while the present study examined population activity of TS without cell type specificity during fear conditioning, Kintshcher et al. examined single unit activity of D1 and D2 neurons during fear conditioning. Authors should carefully discuss previous findings, and indicate consistency, difference and novelty.

4. Authors conclude that dopamine in TS enhances association of CS-US in fear conditioning. However, it is not clear whether these neurons really aid association of two external information or dopamine provides threat information (via threat prediction error) so that cue-threat association was formed. Because the behavioral readout in this study is only freezing, it is hard to dissociate these two possibilities. Optogenetic activation without electric shocks may help to answer.

5. Optogenetic manipulation has to be verified by neural recording. Because some of the stimulation effects could be caused only by supra-physiological activation (Coddington et al., 2023; Long et al. and Masmanidis, 2023), it is important to titrate stimulation parameters by recording activity of the target neurons with the optogenetic manipulation. What is the reason to use 3s optogenetic stimulation rather than 1s for “stimulation at US”? Because 3s stimulation overlaps with CS, and because authors observed behavioral effects with optogenetic activation during CS, authors cannot conclude with this experiment whether optogenetic activation during US is critical or not.

6. DREDD experiments need proper controls. Because CNO may have side effects, authors need to inject CNO in appropriate control animals and compare effects between treatment animals and control animals.

7. Authors found several new insights in the mechanism of fear conditioning via TS. First, dopamine activation in TS during cue facilitates fear conditioning. Second, not only dopamine but also TS neurons exhibit prediction error-like activity pattern. Third, excitation of TS neurons at US (but see Concern 5) enhances fear learning. These are interesting and should be discussed more. For example, dopamine’s role at cue is similar to previous studies in TS (Chen et al., 2022) and in other brain area (Morren et al., 2020), and consistent with the previous proposal in Menegas et al., 2017 that TS dopamine may enhance “CS associability”. It is also important to discuss how population activity of TS in the present study is explained by single neuron activity of D1 vs D2 neurons reported by Kintshcher et al.

Other points

- Kintshcher et al., 2021 found that activity of many TS neurons is locked with movement onset during fear conditioning. Authors should examine relationship between their recording data and movement and discuss the results.
- Pairs should be indicated with lines in case of paired analyses such as Figure 1I and others.
- Decrease of US responses in Figure 1I and 6F is subtle. Authors should be careful for normalization. Because some US responses are sustained, and authors calculated dF/F in each trial using pre-cue activity, the subtle changes may be caused by such normalization. Check the time-course of pre-cue activity without normalization.
- Page 9: why can authors specify “acquisition” by ablation experiments?
- Page 11 line 1: The author referred to this task as a “Pavlovian reward learning task”, but this is an operant task.
- Figure 8: authors observed that significant difference in US responses in the control animals and no significant difference in the 6-OHDA animals, and then conclude that dopamine is necessary for decrease of US responses. This is not the right way of statistical tests; significance cannot be compared. Authors need to directly compare difference in the control vs in the 6-OHDA animals.
- The use of word “sufficient” at multiple locations is confusing and unnecessary when both TS neurons and dopamine are required.

Version 1:

Reviewer comments:

Reviewer #1

(Remarks to the Author)

The authors have performed additional experiments and addressed my previous concerns. This is an important study that shows a critical role of dopamine in the tail of the striatum in a classic fear conditioning paradigm.

Reviewer #2

(Remarks to the Author)

The authors have addressed all my concerns, and I am pleased to see they have conducted extensive new experiments to do so, including loss of function optogenetic assays and also anxiety tasks.

Reviewer #3

(Remarks to the Author)

This revised manuscript by Zafiri is comprehensive in its assessment of dopamine prediction error encoding within SNc projections to tail of the striatum for associative fear learning. The study incorporates numerous cutting-edge techniques and addresses a fundamental question relating to dopamine in this context. The authors have adequately addressed prior reviewer concerns and provided numerous additional experiments and analyses that have significantly strengthened their study. I have no additional comments or concerns.

Reviewer #4

(Remarks to the Author)

The authors did a great job to address all previous concerns. This is a great set of data to advance our knowledge of the tail of the striatum.

Response to Reviewer Comments:

Author's Response: We are grateful for the meticulous and detailed evaluation of our manuscript by the four reviewers. In response to their comments and suggestions, we have conducted 18 new experiments and 7 new analyses of our original data and have also revised the description and discussion of our results. Changes to the manuscript are highlighted in yellow in the revised version of our manuscript. We again thank the reviewers for their thoughtful and constructive comments, which have significantly improved the quality of our manuscript.

Reviewer #1 (Remarks to the Author):

In this study, Zafiri and colleagues examined the role of dopamine neurons projecting to the posterior tail of the striatum (TS) in Pavlovian fear conditioning. The authors examined the activity patterns of TS-projecting dopamine neurons as well as dopamine release in TS, and showed that dopamine inputs to the TS encode prediction error signals for fear learning (threat prediction error or aversive prediction error). Furthermore, suppression and enhancement of dopamine signals in TS reduced and facilitated fear learning. The authors also show that the population activity of TS neurons exhibited prediction error-like activity, which depended on intact dopamine inputs to the TS. Together this study demonstrates that prediction error-like dopamine signals for threat in the TS play a critical role in Pavlovian fear conditioning.

Pavlovian fear conditioning has been studied extensively, yet previous studies focused mainly on the amygdala. This study, together with other recent studies (Kintscher et al., 2023; Chen et al., 2023), identifies the TS as a novel brain area involved in fear conditioning. It is important that the authors show that plastic change in neuronal activity occurs in a dopamine-dependent manner and identify a critical neuromodulatory input that appears to convey prediction error-like signals for fear learning. The results complement recent studies on TS, highlighting its role in avoidance learning. This study overall presents important results, and the results are presented clearly. The conclusions are supported by proper control experiments (e.g. unpaired control, reward learning etc.), yet some experiments lack important controls. These control experiments should be performed to make the results of these experiments interpretable.

Author's Response: We are grateful for the reviewer's very careful and detailed evaluation of our manuscript and thank her/him for constructive comments and helpful suggestions, which have significantly strengthened our study.

1. In fiber photometry experiments, movements of the animal cause artifactual signals. To control for this, the authors need to perform some control experiments such as monitoring signals from control animals expressing EGFP/mCherry or monitoring at the isosbestic wavelength.

Author's Response: The reviewer raises an important concern regarding the fiber photometry recordings. It is indeed important to demonstrate that the observed changes in fluorescence are specifically related to neuronal activity rather than caused by movement artifacts or nonspecific

effects independent of biosensor activity. To address this concern, we have added control groups for our photometry recordings in which we expressed the control fluorophore EYFP in DA neurons or TS neurons and recorded the fluorescence in the DA terminals in the TS or in the TS, respectively. The recordings obtained from these EYFP-expressing mice did not show the transient fluctuations in fluorescence that were observed in the GCaMP-expressing animals (compare Figure 1d to Supplementary Figure 2b, d). Importantly, we did not observe any change in fluorescence around the CS and the US presentations throughout the behavioral protocol in the EYFP-expressing mice (Supplementary Figure 2e), indicating that the responses we observed in GCaMP-expressing mice were not fluorescence changes due to movement artifacts but rather reflected neuronal activity. These results are now presented on Pages 6 and 16 and in Supplementary Figure 2 in the revised manuscript. We also want to point out that these findings are consistent with control fluorophore recordings we previously performed at the cell body level in the VTA as well as at DA terminals in the NAc (Salinas-Hernández et al., 2018; 2023).

2. In experiments using DREADD (pharmacogenetics), the injection of clozapine-N-oxide (CNO) itself might have effects on behavior. The authors present only control experiments using saline injections, and this does not address this potential confound. The authors need to perform control experiments injecting CNO into animals injected with a control virus.

Author's Response: The reviewer is pointing out a valid concern regarding the absence of a CNO injected control group in the original manuscript. We thank the reviewer for bringing this control experiment to our attention, as it is indeed essential to address whether the systemic administration of CNO can affect behavior independently of DREADDs expression. To address this concern, we have now conducted a new control experiment in which we only expressed the fluorophore mCherry instead of DREADDs in TS and injected animals with either CNO or saline prior to fear conditioning. Injection of CNO in these mCherry-expressing control mice did not have a significant effect on fear learning compared to saline injection, suggesting that the observed effect in the chemogenetic experiment was due to activation of the inhibitory DREADDs rather than a nonspecific effect of CNO. The results of this control experiment are presented on Page 17 and in Supplementary Figure 11 in the revised manuscript.

3. Page 10. Line 14. "we hypothesized that DA projections to TS might be important for associating specifically the sensory cues with aversive outcomes. To address this, we next investigated whether TS-projecting DA neurons were necessary for learning the association between the context and the aversive US". A context is ultimately determined by sensory inputs. The dichotomy between sensory cues versus context, therefore, appears to be ill-defined. For example, the presence of long-lasting olfactory cues might have rescued from the effect of 6-OHDA. The conclusions that can be made from this experiment, therefore, need to be discussed more carefully.

Author's Response: The reviewer is raising an interesting point. He/she is correct that a context is ultimately determined by sensory inputs. Indeed, persistent sensory stimuli, such as long-lasting olfactory cues and ambient noise, serve as contextual background elements that define the context during fear conditioning. However, these background sensory stimuli are diffuse and are not temporally and specifically paired with the aversive US, like the CS. During contextual fear conditioning, these diffuse sensory inputs collectively act as the context and the association between the context and the aversive US is formed. On the other hand, in cued fear conditioning, a specific, commonly a brief sensory cue such as a tone is temporally paired with the aversive US. This cue, independent of the background context, comes to predict the exact

temporal occurrence of the US. The distinction between cued versus contextual fear conditioning is well-established in the fear conditioning field and these two different forms of fear learning have been shown to involve distinct neuronal circuits (Maren et al., 2013). For example, the hippocampus and the bed nucleus of the stria terminalis (BNST) are crucial brain structures for contextual but not cued fear conditioning. In our manuscript, we used well-established and standard behavioral protocols for cued versus contextual fear conditioning as has commonly been used in the fear conditioning literature. We have now clarified the distinction between cued and contextual fear learning on Page 11 of the revised manuscript.

4. Page 18, Line 18. The area identified by de Jong et al. (2019) is the ventromedial part of NAc. This study also identified the lateral NAc as encoding canonical reward-related signals. It seems that the other two studies cited here (Yuan et al., 2019; Salinas-Hernandez et al., 2022) found slightly different, more complex distribution of responses. Is there a consensus or is the description used here (“lateral and ventral”) a good summary of those dopamine inputs activated by both rewarding and aversive stimuli?

Author's Response: The reviewer is raising a valid question. de Jong et al. (2019) study indeed reported that while dopamine projections to the ventromedial part of the NAc are activated in response to both aversive and rewarding stimuli, projections to lateral NAc exhibited the canonical reward prediction error-like activity pattern. More recent studies, however, have found activation also by aversive stimuli in DA projections to the lateral NAc (Yuan et al., 2019; Salinas-Hernandez et al., 2023). For instance, Yuan et al. (2019) observed activation to aversive stimuli particularly in the ventral part of the lateral NAc. On the other hand, in Salinas-Hernandez et al. (2023), where recordings were performed across the entire antero-posterior extent of NAc, activation to aversive stimuli was observed more broadly within the lateral NAc and also the ventral part of NAc. The reasons for these varying results across studies remain unclear, and additional research is needed to clarify this. Nevertheless, it is also important to note that earlier studies examining the activity of lateral NAc-projecting DA neurons at the soma level have reported salience-like activity pattern (Lammel et al., 2011). Cai et al. (2020) also found salience coding in DA neurons located in the lateral VTA where the majority of lateral NAc-projecting DA neurons originate. However, given the conflicting results observed for lateral NAc and the need for further research to clarify these findings, we have decided to refer only to ventral, but not lateral, NAc in the revised manuscript. We again thank the reviewer for bringing this point to our attention.

Reviewer #2 (Remarks to the Author):

The authors show that activity in a population of dopaminergic neurons projecting to the tail of the striatum is required for cued, but not contextual fear learning. The authors also show that this circuit is not required for cued-reward learning or for retrieval of cued fear. This work also elegantly shows that the tail of the striatum shows aversion prediction error that requires this dopaminergic innervation. The calcium imaging data demonstrating the mechanisms behind the prediction error signal are well-done and informative. These data are timely and are of interest to the readership of the journal. However, there are important issues that must be addressed, as explained below.

Author's Response: We are grateful for the reviewer's meticulous and thorough review of our manuscript and greatly appreciate his/her valuable comments and suggestions, which have improved the quality of our work.

Major Points

1. Please write the correlation r and p values for all scatterplots in all figures, such as Figure 2E, for example.

Author's Response: We thank the reviewer for bringing this point to our attention. In the manuscript, we used the scatterplots with a unity line to show the distribution of two different variables (e.g., CS responses during habituation versus fear recall in Figure 1k) as paired data points. The goal of these plots was to facilitate visualization of the distribution between the two variables. The unity line, representing the line of equality, illustrates where the values of both variables are identical. The positioning of data points relative to this line indicates which variable has a larger value. For instance, if most points lie below the unity line, as in Figure 1k, it implies that CS responses during fear recall (x-axis) are greater than those during habituation (y-axis). Since our aim was not to assess the correlation between these variables, we did not include correlation r and p values. However, the reviewer raises a valid point that the correlation between these two variables can also be reported in these figures. As requested by the reviewer, we evaluated the correlation between the variables shown in our scatterplots. However, most analyses did not reveal significant correlations. For instance, in Figure 1k, where we plotted CS responses during fear recall versus habituation, we found Pearson's $r = 0.48$ with a p -value of 0.09. Similarly, in Figure 2e, showing CS responses during habituation versus testing in unpaired training, Pearson's $r = 0.15$ with a p -value of 0.74. These results suggest that CS response magnitude in DA neurons projecting to TS during habituation is not predictive of the response magnitude after fear learning or unpaired training. This lack of correlation is not surprising, given the generally low CS responses observed during habituation prior to fear learning. On the other hand, in Figure 2h, which shows predicted versus unpredicted US responses, we observed a strong correlation (Pearson's $r = 0.99$, $p < 0.0001$), which was expected given that US responses were tested in the same session, and their magnitude scaled with predictability. Lastly, in Figure 2l, where responses to footshock US were compared to reward, Pearson's $r = 0.31$ with a p -value of 0.17, indicating no significant correlation between responses to rewards and aversive outcomes in DA projections to TS. Since most comparisons did not reveal significant correlations, and our primary goal in using scatterplots was to facilitate visualization of data rather than highlight correlations, we prefer not to include the correlation results in the manuscript. We feel that presenting these correlation values would detract from the original purpose of the unity line scatterplots, which was to display the distribution of responses between two paired variables. We hope that this is acceptable to the reviewer.

2. Figure 5 shows that optogenetic activation of dopaminergic terminals in TS paired with US or CS increase fear learning. Please add a control showing the effect of activating these terminals during five 30-second epochs outside of CS or US, to demonstrate if the effect shown in 5G and 5H is temporally specific.

Author's Response: We thank the reviewer for bringing up this valid concern regarding the temporal specificity of the effects observed in our experiments involving optogenetic activation of DA terminals in TS. It is indeed important to demonstrate that optogenetic activation of DA terminals in TS per se did not result in a nonspecific increase in freezing levels, and that the behavioral effect was dependent on the temporally precise excitation of the DA terminals during the CS or the US. To address this concern, we have added a control group in which we activated DA terminals in TS for 30 sec during the inter-trial intervals (ITI control) and found that this manipulation did not have a significant effect and ITI control group behaved comparable to

the EYFP controls, suggesting that the effects observed on fear learning were dependent on the temporally-precise activation of DA terminals at the time of the CS or the US. The results of this control experiment are presented on Page 14 and in Figures 5h, 5k and 5l in the revised manuscript.

3. The authors state that temporally precise signaling of the TS is important for fear learning. It is thus puzzling why Figure 7A-F is done with chemogenetics, which is a method that has very low temporal specificity. Please replicate these results using inhibitory optogenetics instead.

Author's Response: The reviewer is correct to point out that we used chemogenetic inhibition rather than optogenetic inhibition in Figure 7 of the original manuscript. At the time of this experiment, whether TS is a crucial structure for fear conditioning was largely unknown. Our first aim was therefore to demonstrate that the activity in TS is indeed required for fear learning and utilized chemogenetic inhibition of TS throughout the fear conditioning session to address this question. The results of the chemogenetic experiment demonstrated that TS is crucial for fear learning. We next performed optogenetic activation of TS neurons at the time of the US to test whether the prediction error-like activity we observed in TS neurons drives fear learning. As the optogenetic activation of TS neurons was done in a temporally-precise manner and provided temporal specificity for the role of TS in driving associative fear learning, we did not additionally perform an optogenetic inhibition experiment in the original manuscript. However, we agree with the reviewer that it is important to also demonstrate that activity in TS is necessary for fear learning in a temporally-specific fashion. To address this, we have now included a new experiment using optogenetic inhibition of TS neurons. In this new experiment, we demonstrate that temporally-precise optogenetic inhibition of TS neurons at the time of the US impairs fear conditioning, indicating that the activity of TS neurons at the time of the US is crucial for driving associative fear learning. We thank the reviewer for drawing our attention to this experiment which helped to strengthen our study. The results of this new experiment are now presented on Pages 17-18 and in Figure 8 of the revised manuscript.

4. Please show if optical inhibition of dopaminergic fibers in TS paired with either US or CS impairs fear learning.

Author's Response: The reviewer is raising a concern similar to his/her previous comment. To demonstrate the requirement of TS-projecting DA neurons in fear learning we performed 6-OHDA ablation of these DA neurons in the original manuscript. However, we acknowledge that this method lacks temporal specificity, and agree that it is important to demonstrate the role of TS-projecting DA neurons in a temporally specific manner. To address this, we have conducted a new experiment using optogenetic inhibition of DA terminals in TS. This experiment shows that temporally-precise optogenetic inhibition of DA terminals in TS during the US significantly impairs fear conditioning, indicating that DA input to TS at the time of the US is crucial for driving associative fear learning. We again thank the reviewer for bringing this experiment to our attention which made our study stronger. The results of this new experiment are now presented on Pages 15-16 and in Figure 6 of the revised manuscript.

5. Please add text to the discussion explaining how and if the circuit studied here is connected to the canonical amygdala fear learning circuitry. Do the authors propose that this circuit affects the amygdala or that it acts independently of the amygdala to modulate fear learning?

Author's Response: We thank the reviewer for bringing this point to our attention. Whether and how the nigrostriatal circuit that we study here is connected to the canonical amygdala fear learning circuitry is an important question. The TS receives strong input from the basolateral amygdala (BLA; Hunnicutt et al., 2016). However, whether these amygdala projections to TS are critical for fear learning is largely unknown. Notably, BLA input to the posterior striatum have been shown to facilitate plasticity in corticostriatal synapses (Popescu et al., 2007), suggesting that amygdala input may be necessary for the plasticity underlying fear acquisition in TS neurons. Future studies investigating the role of amygdala input to TS during fear acquisition will be crucial to address this question. Conversely, although TS sends only sparse projections to the amygdala these projections have recently been shown to be important for fear retrieval (Chen et al., 2023), suggesting a direct connection to the canonical amygdala fear circuitry in mediating fear learning. Further research on how TS input affects neuronal activity in distinct components of the amygdala micro-circuitry will be crucial for elucidating the role of TS in fear retrieval. In addition, the role of TS projections to downstream structures outside the amygdala circuitry during fear acquisition has remained elusive. We have now discussed these points on Pages 26-27 of the revised manuscript.

6. Please show if the major perturbational experiments done also affect overall speed and anxiety. Anxiety (but not speed) data was shown in Figure 5. The potential effects of other manipulations on anxiety and speed were not shown for other experiments controlling neural activity, such as those in Figure 7. As the authors are aware, changes in overall anxiety and speed can be major confounds when the main metric plotted is freezing.

Author's Response: The reviewer has rightly pointed out an important concern. Whether the manipulations we performed (such as optogenetic and chemogenetic manipulations) impact overall speed and anxiety levels is a critical question, as the primary behavioral measure we use is freezing, and any effects on speed and anxiety could potentially confound our results. To address this concern, we have conducted analyses of speed and anxiety-related behaviors using open field and elevated plus maze tests during our optogenetic and chemogenetic experiments. Our findings demonstrate that none of these manipulations significantly affected speed or anxiety levels, suggesting that the results of our perturbation experiments were not confounded by these factors. These new results are now presented on Pages 14-16 and 18-19, and in Figures 5, 6 and Supplementary Figure 14 of the revised manuscript.

Reviewer #3 (Remarks to the Author):

In this study by Zafiri et al., the authors investigate the role of SNc dopamine projections to the tail of the striatum for regulating associative fear learning. In general, the studies are well designed, well controlled, and well executed and provide novel information on the role of dopamine in this specific context in associative fear learning. Although my enthusiasm for the results of the study is generally high, I have some major concerns relating to the interpretation of some results and the overall contextual framing of the manuscript.

Author's Response: We are grateful for the reviewer's very careful and detailed reading of our manuscript and for his/her thoughtful and constructive comments, which have significantly improved the quality of our manuscript.

Specific comments:

1. My biggest concern and disappointment related to this study is the following statement: “although these DA neurons have been implicated in aversion their role in driving fear learning has remained elusive” (lines 19-20). This statement is false. Either the authors are unfamiliar with the literature or have attempted to build a false narrative that borders on the unethical. Aside from the few studies referenced by the authors that relate mostly to fear extinction, numerous lesion studies, pharmacological studies, and electrophysiological studies that have implicated the midbrain dopamine system in fear learning (see Lamont and Kokkinidis, 1998; Nader and LeDoux, 1999; Guarraci and Kapp, 1999; Greba and Kokkinidis 2000, Greba et al., 2001 just to name a few from 100s) there have also been numerous studies directly implicating midbrain dopamine neurons in associative fear learning and the encoding of fear related information during associative learning. Fadok et al., 2009 demonstrated that dopamine production is essential for associative fear learning, and further showed that dopamine production in projection to the amygdala and striatum are important (Fadok et al., 2010 and 2011). Electrophysiological responses and calcium dynamics in response to aversive and fear-inducing USs and the acquisition of CS responding across multiple paradigms have also been reported (Chiodo et al. 1979; Schultz and Romo 1987; Mantz et al. 1989; Ungless et al. 2004; Brischoux et al. 2009; Mirenowicz and Schultz 1996; Guarraci and Kapp 1999; Joshua et al. 2008; Matsumoto and Hikosaka 2009; Wang and Tsien 2011; Zweifel et al. 2011; Cohen et al. 2012; Gore et al., 2014; Luo et al., 2018; Salinas-Hernandez et al., 2018; Jo et al., 2018; Steinberg et al., 2020; and Cai et al., 2020), as have studies directly measuring transient dopamine release during fear-related learning (Roitman et al. 2008 and Badrinarayan et al. 2012). At the functional level, temporally precise inhibition of VTA dopamine neurons at the onset of a CS prevents discriminatory fear learning, as does inhibition of dopamine terminals in the CeA (Jo et al., 2018). Inhibition of SNc neurons projecting the CeA also significantly attenuates fear learning (Steinberg et al., 2020). The authors need to frame their study in the context of these prior studies on the role of dopamine neurons and associative fear learning, not just their own work and literature reviews. In my opinion, the findings presented here make several novel contributions to the field relating to the role of the tail of the striatum specifically; however, it does not benefit them, and is a major disservice to the field and those that have worked hard to advance this field, to claim to have resolved a previously “elusive” function of dopamine in fear learning. They are not the first to implicate dopamine in very specific aspects of associative fear learning, nor are they the first to provide a detailed analysis of the role of dopamine in the encoding of a conditioned fear stimulus and the neuroanatomical function of dopamine circuitry in learning the relationship between a CS and a fear US, give me a break.

Author’s Response: We are grateful to the reviewer for bringing this point to our attention and regret any unintended exclusion or misrepresentation of the previous findings regarding the role of dopamine in fear learning in our original manuscript. The reviewer is correct in highlighting the extensive literature on the role of dopamine’s involvement in fear learning. Previous studies using lesioning, pharmacological manipulations, recordings of DA neuronal activity, measurements of DA release as well as research on DA-deficient mice have consistently demonstrated the critical role of the dopamine system in fear learning and memory. However, it has remained elusive whether ventral midbrain dopamine neurons, located in the VTA and substantia nigra, encode a prediction error signal for aversive outcomes that is crucial for driving associative fear learning. In the sentence cited by the reviewer, we were specifically referring to this open question regarding prediction error signaling as being “elusive,” and did not intend to imply that the role of DA in fear learning in general remains elusive. The third paragraph of the

Introduction, which the reviewer is referring to, specifically focused on the role of DA neurons in prediction error signaling. It is well-established that ventral midbrain DA neurons encode reward prediction errors to drive reinforcement learning (Bayer and Glimcher, 2005; Eshel et al., 2015; 2016; Schultz et al., 1997; Steinberg et al., 2013). Recent studies have also indicated that these DA neurons encode positive prediction errors for the omission of aversive outcomes, which is necessary for driving fear extinction learning (Cai et al., 2020; Luo et al., 2018; Salinas-Hernández et al., 2018; 2023). However, whether these ventral midbrain DA neurons also encode prediction errors for aversive outcomes necessary for driving fear learning is still not known. In this paragraph, we intended to address this outstanding question. However, we realize that we did not explicitly mention ‘prediction error signaling’ in the sentence cited by the reviewer and should have made this point clearer. We have revised this section of the Introduction to convey this outstanding question more clearly. In the revised version we now state: “However, although ventral midbrain DA neurons have been shown to play critical roles in aversion and fear learning and memory (Likhnik and Johansen, 2019; Pezze and Feldon, 2004; Verharen et al., 2020; Zafiri and Duvarci, 2022), whether they contribute to PE signaling necessary for driving fear learning has remained elusive. Importantly, whether DA neurons projecting to brain structures outside the canonical amygdala circuitry encode a PE signal that is crucial for driving associative fear learning is largely unknown.” Furthermore, we acknowledge the reviewer's concern about the need for a more detailed discussion of prior studies on the role of the dopamine system in fear learning. We therefore have included a section on Pages 21-22 of the revised manuscript to frame and discuss our study in the context of this existing literature.

2. The authors repeatedly refer to the observed responses that diminish to the US and increase to the CS as prediction error like. This is only half the equation the authors need to do random omissions to assess prediction errors.

Author's Response: The reviewer is raising an important point. It is well established that dopamine neurons that encode the canonical reward prediction error (RPE) signal exhibit phasic inhibition when an expected reward is randomly omitted. In our study, we observed a prediction error-like activity pattern in the responses to the US and the CS during fear learning. However, whether TS-projecting DA neurons exhibit significant responses to the omission of an aversive US, particularly when a noxious stimulus such as a foot shock is used, is unknown. We therefore agree with the reviewer that examining the activity of these DA neurons during the random omissions of the aversive stimulus is indeed important. To address this, we have conducted a new experiment where the animals underwent a partial fear conditioning task in which the aversive US was omitted randomly in half of the trials (i.e. 5 CS-US and 5 CS alone presentations, see Supplementary Figure 3a). Notably, we found no significant change in the activity of these DA neurons in response to random US omissions. These results are now presented on Page 8 and in Supplementary Figure 3 of the revised manuscript. Likewise, a US omission response was also not seen during the Fear Recall test (see Figure 1g), in which the expected US was omitted at the end of the CS. Importantly, these results are consistent with previous studies that also reported no significant response to US omissions in TS-projecting DA neurons when a mildly aversive air puff was used as the US (Menegas et al., 2017). Collectively, these results suggest that TS-projecting DA neurons do not respond to the omission of an aversive US, indicating a lack of negative prediction error signaling. We have now discussed that TS-projecting DA neurons exhibit only positive prediction errors and lack negative prediction error signaling for aversive stimuli on P. 23 of the revised manuscript. Interestingly, DA neurons located in the dorsal raphe/periaqueductal gray (DR/PAG) that project to central amygdala have been shown to encode prediction error signals during fear learning, yet these DA neurons also do not respond to the omission of aversive USs (Groessl et al., 2018). This finding further supports the notion of a general absence of negative prediction error

signaling in DA neurons encoding aversive prediction errors. Overall, these findings suggest that DA prediction error signaling mechanisms for rewarding versus aversive outcomes likely differ from each other, with DA neurons encoding an aversive prediction error exhibiting responses that are positively shifted and lacking inhibition in response to the omission of aversive stimuli.

3. It is unclear based on the description of the design of the 50% reward probability how this is a measure of uncertainty that would be predicted to cause a change in dopamine release. Where the mice conditioned for numerous days at 90% probability then switched to 50% probability with a randomized design. If they had one day of 90% training then switched to 50% probability with every other nose poke resulting in reward delivery then this is just FR2, not an assessment of uncertainty. It is also not ideal to perform appetitive conditioning on mice after fear conditioning as this diminishes their overall hedonic state.

Author's Response: We thank the reviewer for bringing this point to our attention. We realize that the description of the reward task in the original manuscript was not adequately detailed. We have now described the reward task in more detail in the revised version of the manuscript. Briefly, we used a relatively simple reward task that involved animals entering a nose port and receiving a reward (10% sucrose solution). During a single day of training with a 90% reward probability, the animals quickly learned the task well. We therefore did not give more days of training with 90% probability. During 90% reward probability, we did not record the neuronal activity in the original manuscript. The next day, neuronal activity was recorded while rewards were delivered randomly at a 50% probability. It is important to clarify that the rewarded and unrewarded trials were presented randomly, and we did not use a fixed-ratio 2 (FR2) schedule, as mentioned by the reviewer, which would have required every other nose poke to result in reward delivery. In our task, nose pokes led to reward delivery in an unpredictable, random manner and hence the animals could not predict which nose poke was to be rewarded. Since DA neurons are known to respond strongly when a reward is unpredicted, we utilized this reward task that delivers rewards in an unpredictable fashion. It is important to note that previous studies demonstrated strong reward responses in DA neurons when rewards were delivered with 50% probability (Matsumoto and Hikosaka, 2009). Nevertheless, it is possible that the task that we used did not allow us to measure the maximal reward responses, as it didn't have the highest unpredictability. A similar concern was also raised by Reviewer 4 (comment #2) who stated that **“while the “unpredicted shock” was almost unpredicted, the “unpredicted reward” in Figure 2I is 50% predicted. Therefore, the response to 50% predicted reward in this experiment is likely to be not the maximum reward response of dopamine in TS.”** To address these concerns, we have performed a new experiment in which we performed recordings on the first day of training. As the first reward that the animals receive on the first day has the highest unpredictability and uncertainty, we examined the DA terminal responses to this first reward. Notably, we again observed weak responses that were comparable to the magnitude of responses during 50% reward probability the next day ($n = 7$; $p = 0.93$, signed-rank test; see Figure A below). These results confirm our original findings and indicate that TS-projecting DA neurons respond only weakly to rewards, consistent with previous findings (Menegas et al., 2017; 2018). Because the response magnitudes were not different between the first reward and the average of 50% probability reward, we prefer not to present these new results in the manuscript. We hope this is acceptable to the reviewer.

The reviewer is also raising an important point that prior fear conditioning can reduce an animal's overall hedonic state and hence can affect the responses to rewards when the reward task is performed after fear conditioning. In our study, because our main objective was to investigate the role of DA projections to TS in associative fear learning, we performed the fear conditioning experiments before the reward task. However, we appreciate the reviewer's concern and have conducted a new experiment to evaluate and compare reward responses before and after fear conditioning. Importantly, our results revealed no significant difference in reward responses across these two conditions ($P = 0.37$, signed-rank test; see Figure B below), suggesting that prior fear conditioning did not affect the reward response magnitude in DA neurons projecting to TS. The results of this experiment are now presented on Pages 34-35 of the revised manuscript.

4. Assessment of motivational salience requires varying the magnitude of the unconditioned stimulus and assessing the relationship between the observed signals, the conditioned behavioral responses, and the magnitude of the US.

Author's Response: The reviewer has raised a valid point. He/she is correct in noting that evaluating motivational salience also requires varying the magnitude of the unconditioned stimulus and analyzing both neuronal and behavioral responses in relation to this variation. Since we did not conduct this experiment, we have now revised the Results section and removed any reference to 'motivational salience' in the revised manuscript.

5. The tail of the striatum contains D1 and D2 receptor expressing neurons. Presumably these neurons respond differently to dopamine (stimulation versus inhibition); thus, it is unclear how bulk stimulation or inhibition of all TS neurons influences associative learning. This is not a major criticism, but the caveat should be addressed.

Author's Response: The reviewer is raising an important issue. He/she is right to point out that TS contains different types of neurons such as D1- and D2-receptor expressing medium spiny neurons (D1- and D2-MSNs) and that they are expected to respond differently to the DA input. In our chemogenetic and optogenetic experiments, we performed stimulation or inhibition of all TS neurons and found that these manipulations enhanced and impaired associative fear learning, respectively. Although the effects of bulk manipulation of all TS neurons on fear learning may seem surprising, it is important to note that similar results were also obtained in Chen et al. (2023) where optogenetic inhibition of all TS neurons caused an impairment in auditory fear conditioning. It is also important to emphasize that similar results are also observed in the amygdala. For instance, although the basolateral amygdala (BLA) contains different kinds of neurons with opposing roles in fear learning versus fear extinction and reward, bulk inhibition of all BLA neurons (e.g. using reversible inactivation with muscimol which is similar to the chemogenetic inhibition we used in our study) causes impairments in fear learning and memory (e.g. Amano et al., 2011; Herry et al., 2008; Sierra-Mercado et al., 2011; Wilensky et al., 1999). Our results are therefore in line with previous studies that have inhibited the activity of all neurons within TS or the amygdala and had a significant effect on fear learning. The distinct roles D1- and D2-MSNs of TS play in fear learning are not yet fully understood. Kintscher et al. (2023) investigated the role of these neurons in the ventralmost part of the TS, largely including the amygdalostratial transition area (ASt), and found that while optogenetic inhibition of D1-MSNs impaired fear retrieval, inhibition of D2-MSNs had no significant effect. Furthermore, inhibiting D1- and D2-MSNs in this study had no effect on movement or any overt behavior, which is consistent with our own findings that TS inhibition did not affect movement or anxiety-like behaviors (see Supplementary Figure 14). The role of D1- and D2-MSNs in fear learning across the larger, more dorsal TS remains an open question for future research. If D1-MSNs, but not D2-MSNs, are similarly critical in the more dorsal TS, where our study was focused, it is possible that our optogenetic and chemogenetic manipulations influenced fear learning primarily through their effect on D1-MSNs. We have now discussed this issue in more detail on P.28 of the revised manuscript.

Reviewer #4 (Remarks to the Author):

In the present paper, Zafiri et al. investigated the role of dopamine (DA) neurons projecting to the tail of the striatum (TS) in associative fear learning. First, they used fiber photometry to record from three different indicators – GCaMP in TS-projecting DA neurons, GCaMP in TS neurons, and dopamine sensor in the TS – during associative fear learning (a tone paired with a footshock) and a following fear recall session. They found remarkably similar activity in each case: all showed characteristics of an aversive prediction error (PE) signal. Next, they established that activity of TS-projecting DA neurons and TS neurons are necessary for associative fear learning. Lesioning TS-projecting DA neurons before fear conditioning or chemogenetically inhibiting TS neurons during fear conditioning led to impaired learning during conditioning and recall. Furthermore, optogenetic excitation of TS-projecting DA neurons during the CS or the US, or excitation of TS neurons during the US, strengthened the effect of fear conditioning, enhancing freezing behavior. Furthermore, when TS-projecting DA neurons were lesioned, prediction-error like signaling in TS neurons was abolished, suggesting that the emergence of PE-like activity patterns in TS relies on dopaminergic input. Interestingly, despite its key role in associative fear learning, TS DA was not necessary

for contextual fear learning, as animals with TS DA lesions still showed robust freezing behavior when placed back into the fear conditioning context.

Overall these experiments were logically constructed and comprehensive for addressing the authors' posed questions, including appropriate controls in most parts. The results were also clearly presented. This study's findings overlap considerably with past works (Menegas et al., 2017, 2018), which showed prediction error-like coding in TS DA neurons during classical conditioning and TS DA's role in learning of aversive airpuff, and this diminishes the novelty of the work. However, considering controversies in roles of TS, the beautiful set of results in the present study will contribute to the field. Further, the authors present several useful advances, including clarifying the role of TS DA in associative and contextual fear learning and the specific role in learning but not expression of conditioned fear. The work would be considerably strengthened and rendered more novel with the addition of more detailed analyses of the behavioral paradigms, a few more targeted experiments, and discussion of the novel findings, suggestions for which are contained below along with general issues.

Author's Response: We are grateful for the reviewer's very careful and detailed evaluation of our manuscript and thank her/him for constructive comments and helpful suggestions, which have significantly strengthened our study.

Major concerns

1. The lack of an impairment in the contextual fear learning paradigm following TS DA lesion is interesting. Contextual fear learning depends to some extent upon associative fear learning: animals must at least associate sensory stimuli in the environment with the unconditioned stimulus to develop this fear. One possibility is that TS DA is required for learning to associate auditory stimuli with aversive outcomes, but not stimuli of other sensory modalities. To dissociate auditory vs other modalities, and cued vs contextual learning, authors should perform their fear conditioning experiments with another sensory modality as the CS – for instance, a flash of light.

Author's Response: We thank the reviewer for bringing this issue to our attention. The reviewer is right to suggest that DA input to TS may be required for learning to associate specifically auditory stimuli with aversive outcomes, but not stimuli from other sensory modalities. To address this question, we performed a fear conditioning experiment using a visual CS (i.e. 30s light cue), following the reviewer's suggestion. We found that 6-OHDA ablation of TS-projecting DA neurons impaired acquisition of visual fear conditioning, indicating that these DA neurons are crucial for associating not only auditory but also visual stimuli with aversive outcomes. These results suggest that TS-projecting DA neurons play a critical role in cued fear learning across different sensory modalities, and further support our conclusion that DA in TS is required for cued but not contextual fear learning. The results of this new experiment are presented on Pages 11-12 and in Supplementary Figure 6 of the revised manuscript.

2. The separation between fear conditioning and reward conditioning is interesting. Because it is very important, authors need more careful examination using comparative conditions. For comparison of neural responses, while the "unpredicted shock" is almost unpredicted, the "unpredicted reward" in Figure 2I is 50% predicted. Therefore, the response to 50% predicted reward in this experiment is likely to be not the maximum reward response of dopamine in TS. Moreover, the dopamine response at the reward

timing in this task is difficult to interpret because it includes both the reward response and the nose entry response, which could reflect an increase in reward expectation.

Author's Response: We thank the reviewer for drawing our attention to these issues. The reviewer is raising a valid point that unpredicted shock versus reward responses should be examined using comparative conditions. In our reward task, random delivery of rewards with 50% probability makes the reward unpredictable but it is still more predictable than the unpredicted shock. The reviewer is right to point out that the response to 50% predicted reward may not be the maximum reward response of DA terminals in the TS. To address this concern, we have conducted a new experiment where we recorded the response of DA terminals in TS to the first unpredicted reward that animals received during the first day of training. As this would be the first unpredicted reward that animals receive, it is comparable to the first unpredicted shock. Notably, we again observed weak responses that were comparable to the magnitude of responses during 50% reward probability the next day ($n = 7$; $p = 0.93$, signed-rank test; see Figure A below). Furthermore, responses to the first reward were again smaller compared to the first footshock USs (see Figure B below). These results confirm our original findings and indicate that TS-projecting DA neurons respond only weakly to rewards, consistent with previous studies (Menegas et al., 2017; 2018). Because the magnitude of responses to the first reward on day 1 and the average of 50% rewards on day 2 are comparable, we prefer not to show these new results in the manuscript. We hope this is acceptable to the reviewer.

In addition, in our reward task in freely-behaving animals, the 'reward response' we measured likely includes not only the direct response to reward but also responses to nose port entry and reward expectation, as the reviewer pointed out. We indeed saw a ramping of neuronal activity right before the reward delivery, which likely reflects reward expectation. However, this suggests that the actual reward response may be smaller, rather than larger, than our calculated 'reward response.' Consequently, these factors do not contest our conclusion that TS-projecting DA neurons are only weakly activated by rewards. Furthermore, our results are also consistent with and support previous studies conducted in head-fixed mice which also showed weak reward responses (Menegas et al., 2017; 2018), highlighting the consistency of results across both head-fixed and freely-behaving conditions.

Because authors chose to use operant task for reward while footshock is passive, it is difficult to directly compare. Again, similar to the Concern 1, to dissociate classical vs

instrumental learning, and reward vs threat learning, I recommend that the authors reconsider the task designs to make them comparable. For behavioral effects of dopamine manipulation, it is also important to use similar tasks rather than use passive fear conditioning and operant reward acquisition.

Author's Response: We thank the reviewer for raising concerns about the comparability of the tasks we used to investigate fear versus reward learning. In our manuscript, we used classical fear conditioning to study associative fear learning, where animals received footshocks passively and their innate reflexive fear responses, such as behavioral freezing, served as a measure of fear learning. Because the major objective of our study was to investigate the role of the dopamine-TS circuitry in classical, rather than instrumental, fear learning in freely-behaving mice we used classical fear conditioning. Using a comparable reward task would have necessitated a classical reward learning paradigm, where animals would need to receive rewards passively by direct delivery into the mouth, and the innate reflexive reward responses, such as salivation, need to be used to measure learning. This approach would be practically very difficult to implement and would require additional establishment in our lab, which is not feasible within the time frame of the revisions. In fact, the majority of studies on reward learning in the literature utilize operant tasks, where animals must perform an action in order to receive a reward, and the frequency of that action is used as a measure of learning. Even in head-fixed conditions, where animal movement is restricted, operant reward tasks are used where the animals need to perform the action of licking the reward sprout in order to receive the reward, and the lick rate is indeed used as the measure of learning. In our manuscript, we aimed to use simple reward tasks in freely-behaving animals to enable comparison with fear conditioning in freely-moving mice. To measure reward responses during our recording experiments, the task involved training animals to enter a reward port to receive a reward. In these experiments, the unpredicted reward responses were compared to unpredicted footshock responses. In 6-OHDA ablation experiments, we used a reward learning task similar to cued fear conditioning, in which animals learned a simple CS-reward association, and they had to enter the reward port during the CS in order to receive the reward. While these reward tasks are not directly comparable to classical fear conditioning, we believe they are suitable options for comparisons in freely-behaving mice. We hope this is acceptable to the reviewer. We have now discussed that our fear conditioning and reward learning tasks are not directly comparable on Page 24 of the revised manuscript.

Further, it is also important to analyze behaviors more systematically, instead of relying on only one measurement (whether reward is acquired after cue). For example, the lesioned animals could get the same number of rewards following tone without forming a clear association – they could simply continuously nose-poke into the port, whether or not the tone was playing. The authors should show other behavioral correlates such as number of pokes into the port following the tone and in the inter-tone interval, length of time in the port, and velocity, to investigate such possibilities.

Author's Response: We are grateful for the reviewer for bringing this point to our attention. In the 6-OHDA experiments, the number of rewarded CS presentations was used as a measure of learning in the original manuscript. However, the reviewer is raising a valid point that it would be good to show more behavioral correlates when comparing learning levels between control and lesioned mice. To address this concern, we have now analyzed the latency to enter the port, time spent in the port during the CS and number of pokes into the port in the inter-trial intervals. Notably, we did not find any significant difference when these additional behavioral correlates were compared between the two groups, supporting our conclusion that learning of a simple cue-reward association does not require TS-projecting DA neurons. The results of these

additional analyses are presented on Page 12 and in Supplementary Figure 7 in the revised manuscript.

3. Authors should frame their study in a more accurate way in the fields since many findings are overlapped with findings in previous studies by Menegas et al., 2017, 2018, Kintshcher et al., 2021, and Chen et al., 2023. For example, Menegas et al. found that TS-projecting dopamine neurons play a critical role in learning of aversive air puff, using ablation with 6OHDA. They also found that ablation of TS-projecting dopamine neurons does not affect learning of reward value in exactly the same task as threat learning. The present study did not discuss these although they confirmed these previous findings.

Similarly, although Kintshcher et al. and Chen et al. examined TS roles in fear conditioning, there is very limited discussion of these studies. Especially, while the present study examined population activity of TS without cell type specificity during fear conditioning, Kintshcher et al. examined single unit activity of D1 and D2 neurons during fear conditioning. Authors should carefully discuss previous findings, and indicate consistency, difference and novelty.

Author's Response: We appreciate the reviewer's thoughtful comments and regret that our original manuscript did not offer a more thorough comparison with the studies of Menegas et al. (2017, 2018), Kintshcher et al. (2023), and Chen et al. (2023). While Menegas et al. (2017, 2018) observed a prediction error-like activity in TS-projecting DA neurons during the association of a CS and an airpuff, our study is the first to demonstrate that TS-projecting DA neurons encode a prediction error signal that is crucial for driving associative fear learning. Our findings overlap with those of Menegas et al. in several ways. First, Menegas et al. showed that these DA neurons are activated particularly by novel and external threats such as airpuffs and loud tones but not by bitter taste. In line with this, we show that these neurons are also strongly activated by painful stimuli such as footshocks. However, whether these DA neurons encode prediction errors for strong and painful stimuli such as foot shocks during fear conditioning remained unknown, which we demonstrate in Figure 1 of our manuscript. We further demonstrate that the responses of these DA neurons to the CS emerge as a result of CS-US association during fear conditioning, with unpaired presentations of the CS and US failing to potentiate CS responses (Figure 2). Moreover, we show that these neurons have a larger response to an unpredicted footshock US compared to the predicted US, providing additional evidence for their role in prediction error signaling.

In addition, Menegas et al. (2018) demonstrated the role of TS-projecting DA neurons in novelty-induced threat avoidance. As the reviewer highlighted, this study showed that 6-OHDA ablation of TS-projecting DA neurons impairs choice bias against an aversive airpuff suggesting that these DA neurons are required for learning about and avoiding threats. Our findings are consistent with these results and further extend this by showing that these neurons are crucial for learning the CS-US association during fear conditioning, specifically during the acquisition phase, but not the retrieval or expression phases (see Figure 4). Furthermore, our study is the first to demonstrate that these neurons are essential for associating discrete cues with aversive USs but not for learning cue-reward associations or contextual fear learning, highlighting their specific role in cued associative fear learning.

Finally, in line with Menegas et al. (2018), we find that these DA neurons are not involved in learning cue-reward associations. In the revised manuscript, we include new data (Supplementary Figure 4) showing that changes in reward value do not significantly affect the activity of these DA neurons. Importantly, our study also provides novel evidence that

bidirectional optogenetic manipulation of DA terminals in TS during the CS or US can modulate fear learning, thus advancing our understanding of how these neurons drive associative fear learning. In the revised manuscript, we now provide more detailed discussion of the findings of Menegas et al. (2017, 2018), highlighting the parallels and consistencies between their studies and our findings.

Our study also shares notable similarities with Chen et al. (2023), particularly with respect to the involvement of TS-projecting DA and TS neurons in fear learning. However, our work uniquely focuses on prediction error signaling in these neurons, which is crucial for driving associative fear learning, an aspect not investigated by Chen et al. (2023). There are both consistent and divergent findings between our study and theirs.

For instance, Chen et al. (2023) observed that optogenetic inhibition of DA terminals in TS impaired fear retrieval but not fear acquisition. In contrast, our findings show that both lesioning and optogenetic inhibition of TS-projecting DA neurons impair fear acquisition (see Figures 4 and 6). Similarly, while Chen et al. found that inhibiting TS neurons impaired fear retrieval, we found that inhibiting TS neurons through both chemogenetic and optogenetic methods impaired fear acquisition (see Figure 8). While the reasons for these discrepancies remain unclear, we also demonstrate that optogenetic activation of TS neurons during the US enhances fear learning.

Although Chen et al. (2023) found that fear conditioning enhanced CS responses in TS-projecting DA and TS neurons, their study did not assess neuronal responses to the US. Our study corroborates the finding of CS response potentiation, but we further show that US responses decrease over the course of fear conditioning and that both DA and TS neuronal activity exhibit prediction error signaling that is necessary for driving fear learning.

Lastly, Chen et al. showed that TS-projecting substantia nigra (SN) neurons were important for potentiation of CS responses during fear conditioning. However, whether this was dependent on DA input to TS remained unclear as inhibition of neuronal activity in SN lacked DA neuron specificity. In our study, we demonstrate that ablation of specifically the TS-projecting DA neurons prevented potentiation of CS responses as well as predictive coding of US responses in parallel with impaired fear learning. In summary, although there are several overlaps with Chen et al., 2023 as well as Menegas et al., 2017 and 2018, our study provides novel insights into the role of prediction error signaling in this unique dopaminergic nigrostriatal circuit in driving associative fear learning. We have now included more comprehensive discussion of the findings of Chen et al., 2023 highlighting the consistencies, differences, and novel contributions of our findings in the revised manuscript.

Kintscher et al. (2023) investigated the role of D1 and D2 medium spiny neurons (MSNs) in the ventralmost portion of the TS, which largely includes the amygdalostriatal transition area (ASt). In contrast, our study focused on the more dorsal and larger portion of the TS, intentionally avoiding the ventralmost area due to its proximity to the amygdala and the inclusion of the ASt. Therefore, it is important to note that there is minimal overlap between the region of the TS studied in our research and that targeted by Kintscher et al. (2023). Nevertheless, Kintscher et al. (2023) found that while D1-MSNs exhibited enhanced CS responses as a result of fear conditioning, D2-MSNs did not. Furthermore, optogenetic inhibition of D1-MSNs impaired fear retrieval whereas inhibition of D2-MSNs had no significant effect. This study indicates a critical role for D1-MSNs in the ventralmost TS and the ASt, during fear retrieval. However, the role of D1- and D2-MSNs in associative fear learning across the larger, more dorsal TS, where our study was focused, currently remains an open question. Future research investigating the

activity of D1- and D2- MSNs in the broader more dorsal TS during fear learning will be crucial for addressing this question. We have discussed the findings of Kintscher et al. (2023) in relation to our study on P. 28 of the revised manuscript.

For the neural activity, Menegas et al. found that these dopamine neurons signal threat or physical salience of external stimuli, but not aversiveness per se because the response was selective with salient external stimuli but not with aversive bitter taste. These studies also found that dopamine responses have some characteristics of prediction error, but not perfect prediction error because they did not show inhibitory responses to omission. The present study only examined an electric shock and did not examine responses to other aversive stimuli or omission of aversive stimuli. While results in the present study is consistent with the previous proposal of threat prediction error, it is not justified that these dopamine neurons signal aversiveness or aversive prediction error in contrast to previous studies.

Author's Response: The reviewer is raising a valid concern. Consistent with Menegas et al., our study also showed that TS-projecting DA neurons do not signal aversiveness per se, as optogenetic activation of DA terminals in TS did not induce real-time place aversion (see Figure 5m) or induce fear learning on its own in the absence of the aversive US (see Figures 5f, 5i, 5j). Because we found that these DA neurons encode a prediction error signal for the aversive US during fear learning, we referred to this signal as the 'aversive' prediction error. However, the reviewer is correct to point out that these DA neurons do not respond to all aversive stimuli such as bitter taste and that they have been shown to respond to aversive and salient external stimuli in previous studies (Menegas et al., 2017; 2018). We appreciate the reviewer's concern and have now revised the manuscript to remove the term 'aversive' in reference to the prediction error.

Furthermore, the reviewer also raises an important issue that we did not test the responses of these DA neurons to the omission of the footshock US to examine whether they exhibit negative prediction error signaling. The same issue was also raised by Reviewer 3 (comment #2). To address this, we have conducted a new experiment where the animals underwent a partial reinforcement task in which the aversive US was omitted randomly in half of the trials (see Supplementary Figure 3a). Notably, we found no significant change in the activity of these DA neurons in response to random US omissions. These new results are presented on Page 8 and in Supplementary Figure 3 of the revised manuscript. Our results are consistent with previous studies that also reported no significant response to US omissions in TS-projecting DA neurons when a mildly aversive air puff was used as the US (Menegas et al., 2017). Together, these results suggest that TS-projecting DA neurons do not exhibit inhibition in response to the omission of an aversive US, indicating a lack of negative prediction error signaling. Interestingly, DA neurons located in the dorsal raphe/periaqueductal gray (DR/PAG) that project to central amygdala have been shown to encode prediction error signals during fear learning, yet these DA neurons also do not respond to the omission of the aversive US (Groessl et al., 2018). This finding further suggests a general absence of negative prediction error signaling in DA neurons encoding prediction errors during associative fear learning. Overall, these findings raise the possibility that DA prediction error signaling mechanisms for rewarding versus aversive outcomes might differ from each other, with DA neurons encoding prediction errors for aversive outcomes exhibiting positively shifted responses and lacking inhibitory responses to the omission of aversive stimuli.

4. Authors conclude that dopamine in TS enhances association of CS-US in fear conditioning. However, it is not clear whether these neurons really aid association of two

external information or dopamine provides threat information (via threat prediction error) so that cue-threat association was formed. Because the behavioral readout in this study is only freezing, it is hard to dissociate these two possibilities. Optogenetic activation without electric shocks may help to answer.

Author's Response: The reviewer raises a critical point. It is indeed possible that excitation of DA terminals in TS might be aversive *per se* and could convey threat information independent of the footshock US, as the reviewer suggested. This could lead to learning of an association between the CS and DA terminal excitation, rather than strengthening the CS-footshock US association. To address this, we have added a control group where DA terminals in TS were excited at the end of the CS but without the footshock US (No-US Control). Our findings show that this manipulation did not result in fear learning, indicating that excitation of DA terminals in TS does not induce aversion *per se* or signal threat. Importantly, this finding is consistent with our results of the real-time place preference experiment in the original manuscript where excitation of DA terminals in TS did not induce place avoidance (Figure 5m), also suggesting that this manipulation does not induce aversion *per se*. The results of the No-US control experiment are presented on Page 13 and in Figures 4f, 4i and 4j in the revised manuscript.

5. Optogenetic manipulation has to be verified by neural recording. Because some of the stimulation effects could be caused only by supra-physiological activation (Coddington et al., 2023; Long et al. and Masmanidis, 2023), it is important to titrate stimulation parameters by recording activity of the target neurons with the optogenetic manipulation. What is the reason to use 3s optogenetic stimulation rather than 1s for “stimulation at US”? Because 3s stimulation overlaps with CS, and because authors observed behavioral effects with optogenetic activation during CS, authors cannot conclude with this experiment whether optogenetic activation during US is critical or not.

Author's Response: The reviewer raises an important issue and is correct to point out that some of the optogenetic stimulation effects can be caused by supra-physiological activation of neurons. However, the intensity of light and stimulation parameters that we used in the manuscript are comparable to those commonly used in similar studies in the literature (e.g. see Menegas et al., 2018 Nature Neuro; Kintscher et al., 2023 Nature Commun; de Jong et al., 2019 Neuron; Salinas-Hernández et al., 2018 ELife). Importantly, the stimulation frequency that we used (20 Hz) falls within the physiological firing range of DA neurons (see Cohen et al., 2012) and TS neurons (see Nardoci et al., 2022). Furthermore, in response to Reviewer's concern #4 and the concern #2 of Reviewer 2, in the revised manuscript we have now included No-US and ITI control groups, in which optogenetic excitation is delivered in the absence of the US or during the ITIs. These control manipulations did not cause any behavioral effects, suggesting that our stimulation parameters likely did not result in supra-physiological neuronal activation and likely did not produce non-physiological effects. It is also important to note that optogenetic excitation experiment is not the only experiment supporting our conclusion but optogenetic inhibition of DA terminals that we have conducted during the revisions of our manuscript (as response to Reviewer 2) further support our conclusion that TS-projecting DA neurons encode PEs that act as teaching signals to drive associative fear learning.

The reviewer is also raising a valid concern regarding the duration of optogenetic stimulation during the US. Our original 3s stimulation overlapped with the CS for 1 second. It is therefore possible that the observed behavioral effect was due to activation during the CS, rather than the US, as the reviewer suggests. To address this, we conducted an additional experiment using a 1s optogenetic stimulation at the time of the US and found that 1s excitation also enhanced fear

learning (see Figure below). This indicates that optogenetic activation during the US alone is indeed sufficient to enhance fear learning. Since the results for the 1s and 3s stimulations were statistically comparable, we have combined these data in Figures 4i and 4j of the revised manuscript. Notably, optogenetic inhibition experiments where we optogenetically inhibited DA terminals in TS as well as TS neurons in a temporally-specific manner during the US in the revised manuscript (as response to Reviewer 2) further demonstrate that neuronal activity at the time of the US is critical for driving associative fear learning. These new optogenetic inhibition experiments are now presented in Figures 6 and 8 in the revised manuscript.

6. DREADD experiments need proper controls. Because CNO may have side effects, authors need to inject CNO in appropriate control animals and compare effects between treatment animals and control animals.

Author's Response: The reviewer raises a valid concern regarding the absence of a CNO injected control group in the original manuscript. We thank the reviewer for bringing this control experiment to our attention, as it is indeed essential to address whether the systemic administration of CNO can affect behavior independently of DREADDs expression. To address this, we have conducted a new control experiment in which we expressed only the fluorophore mCherry instead of DREADDs in TS and injected animals with either CNO or saline prior to fear conditioning. Injection of CNO in these mCherry-expressing control mice did not have a significant effect on fear learning compared to saline injection, suggesting that the effect we observed in the chemogenetic experiment was due to activation of the inhibitory DREADDs rather than a nonspecific effect of CNO. The results of this control experiment are presented on Page 17 and in Supplementary Figure 11 in the revised manuscript.

7. Authors found several new insights in the mechanism of fear conditioning via TS. First, dopamine activation in TS during cue facilitates fear conditioning. Second, not only dopamine but also TS neurons exhibit prediction error-like activity pattern. Third, excitation of TS neurons at US (but see Concern 5) enhances fear learning. These are interesting and should be discussed more. For example, dopamine's role at cue is similar to previous studies in TS (Chen et al., 2022) and in other brain area (Morren et al., 2020), and consistent with the previous proposal in Menegas et al., 2017 that TS dopamine may enhance "CS associability". It is also important to discuss how population activity of TS in the present study is explained by single neuron activity of D1 vs D2 neurons reported by Kintshcher et al.

Author's Response: We thank the reviewer for bringing these points to our attention. We regret that we did not emphasize the new insights from our study in the original manuscript. We have now elaborated on the novel aspects of our study in more detail in the revised manuscript, as recommended by the reviewer. We have also included a section in the Discussion on Page 28 where we discuss how population activity of TS can be explained by single neuron activity of D1 vs D2 neurons reported by Kintscher et al. (2023). Kintscher et al. (2023) investigated the role of D1 and D2 neurons in the ventralmost part of the TS, largely including the amygdalostriatal transition area (ASt), and found that while D1 neurons exhibited fear learning-related activity patterns, D2 neurons did not. Furthermore, optogenetic inhibition of D1-neurons impaired fear retrieval whereas inhibition of D2 neurons had no significant effect. However, the role of D1 and D2 neurons in associative fear learning across the larger, more dorsal TS — where our study was focused — currently remains an open question. If D1, but not D2, neurons are similarly involved in fear learning in the more dorsal TS, it is possible that the average population activity observed in our study was mainly driven by the activity of the D1 neurons. Future research investigating the activity of D1 and D2 neurons in the broader more dorsal TS during fear learning will be crucial for addressing this question.

Other points

• **Kintshcher et al., 2021 found that activity of many TS neurons is locked with movement onset during fear conditioning. Authors should examine relationship between their recording data and movement and discuss the results.**

Author's Response: The reviewer has highlighted an important issue. Kintscher et al. (2023) found subpopulations of neurons in the ventral TS that were activated at the onset of movement, but concluded that this movement-ON driven activity did not contribute to the control of movement and further noted that the role of this movement-related activity in the ventral TS remains unknown. In line with the findings of this study, our optogenetic activation of TS neurons also did not affect movement in the open field test (see Supplementary Figure 14d) indicating that these neurons likely are not involved in motor behavior. Given that the major goal of our manuscript is to understand the role of TS neurons in associative fear learning, we focused our analysis on the CS- and US-responsive activity. We agree that the relationship between TS neuronal activity and movement is a critical question. However, addressing this question requires an in-depth analysis of movement behavior and its correlation with neuronal activity, which is beyond the scope of our current manuscript. We plan to address this important question in detail in a future study.

• **Pairs should be indicated with lines in case of paired analyses such as Figure 1I and others.**

Author's Response: We appreciate the reviewer's valuable suggestion. In Figures where paired analyses are presented, we now indicate pairs with lines in the revised manuscript.

• **Decrease of US responses in Figure 1I and 6F is subtle. Authors should be careful for normalization. Because some US responses are sustained, and authors calculated dF/F in each trial using pre-cue activity, the subtle changes may be caused by such normalization. Check the time-course of pre-cue activity without normalization.**

Author's Response: The reviewer is raising an important concern. If US responses were sustained, they could indeed contaminate the baseline, resulting in elevated pre-CS activity and

potentially leading to reduced US responses in the next trial during fear conditioning, given that we calculated dF/F for each trial based on pre-CS activity. However, given the long inter-trial intervals we used (40-120 seconds) and that US responses generally returned to baseline levels within 5 seconds (see Figure 1g), it is unlikely that pre-CS activity was significantly affected by the previous trial's US response. Nevertheless, to address this concern, we analyzed the pre-CS activity without normalization as the reviewer suggested. For GCaMP recordings, we found a slight but significant decrease, rather than an increase, in the baseline activity through the course of the fear conditioning session ($*P = 0.016$, signed-rank test), which likely resulted from photobleaching. In contrast, for dLight recordings, there was no significant difference in the pre-CS baseline activity from the first to the last trial of fear conditioning ($P = 0.1$, signed-rank test), possibly due to differences in photobleaching dynamics between GCaMP and dLight. Notably, the decrease in US responses from the first to the last US trial was stronger in the dLight recordings ($**P = 0.009$, signed-rank test) compared to the GCaMP recordings ($*P = 0.013$, signed-rank test), suggesting that the reduction in US responses occurred independently of any changes in baseline activity. In addition, we observed a similar decrease in baseline GCaMP activity ($*P = 0.04$, signed-rank test) during the reward session. However, there was no significant difference in the magnitude of the reward responses when the first and last reward trials were compared ($P = 0.63$, signed-rank test), suggesting again that the observed reduction in US responses was unlikely due to changes in baseline activity in the GCaMP recordings. Furthermore, we also analyzed the baseline activity for the session where predicted and unpredicted USs were randomly delivered (Figure 2f-h). As these two trial types were presented randomly, the previous trial's US responses would be expected to have similar effects on both types of US responses. Yet, we observed an even greater reduction in responses to the predicted USs when compared to unpredicted USs ($**P = 0.0078$, signed-rank test). Overall, these results indicate that the reduction in US responses that we observed during the course of fear conditioning reflected prediction error coding rather than being an artifact of normalization.

• **Page 9: why can authors specify “acquisition” by ablation experiments?**

Author's Response: The reviewer is raising an important question. In 6-OHDA experiments, we performed ablation of TS-projecting DA neurons at two distinct time points. We first lesioned these neurons before the acquisition phase of fear conditioning (Figure 4c-e), which led to impaired fear learning, indicating that these DA neurons are crucial during fear acquisition. In contrast, when we performed the 6-OHDA ablation 48 hr after fear conditioning (Figure 4f-h), once the CS-US association had been established, we found no effect on the retrieval or expression of fear memories. Therefore the impairment caused when the lesion was performed before acquisition has to be due an effect on acquisition, and not on retrieval or expression. These results together suggest that TS-projecting DA neurons are specifically required during the acquisition of the CS-US association but not for the retrieval or expression of these memories.

• **Page 11 line 1: The author referred to this task as a “Pavlovian reward learning task”, but this is an operant task.**

Author's Response: The reviewer is raising a valid point. Operant reward learning tasks involving a simple cue and reward association have been referred to as Pavlovian reward tasks in the literature (for example see Steinberg et al., 2013 Nature Neuro; Steinberg et al., 2020 Neuron). In line with these previous studies, we used this terminology in our initial submission. However, we agree with the reviewer's concern and have revised the manuscript to remove the reference to our reward task as a 'Pavlovian' task.

• **Figure 8: authors observed that significant difference in US responses in the control animals and no significant difference in the 6-OHDA animals, and then conclude that dopamine is necessary for decrease of US responses. This is not the right way of statistical tests; significance cannot be compared. Authors need to directly compare difference in the control vs in the 6-OHDA animals.**

Author's Response: We thank the reviewer for bringing this issue to our attention. The reviewer is raising a valid point that we need to also directly compare the difference in US responses in the control versus 6-OHDA groups. In our original dataset, while we observed a strong trend, it did not reach statistical significance. This was likely due to the smaller number of animals and recording sites in the control group (n = 11) relative to the 6-OHDA group (n = 14). For the revised manuscript, we have conducted additional recordings, increasing the control group to 14 recording sites to match the 6-OHDA group. The updated analysis now reveals a significant difference between the control and 6-OHDA groups when we compare the difference in US responses as well as CS responses. These results are now presented on Page 20 and in Figure 9n-m.

• **The use of word “sufficient” at multiple locations is confusing and unnecessary when both TS neurons and dopamine are required.**

Author's Response: We appreciate the reviewer's concern and regret any unintended misrepresentation of our results. However, in the manuscript, we did not assert that TS-projecting DA neurons or TS neurons are independently sufficient for fear learning. Instead, we noted that optogenetic activation of these neurons during fear conditioning is 'sufficient' or adequate to enhance associative fear learning. Nevertheless, we acknowledge that using the term 'sufficient' can lead to misunderstandings. We have therefore removed it in the revised version of the manuscript.

REFERENCES

Amano, T., Duvarci, S., Popa, D., Paré, D. (2011). The fear circuit revisited: contributions of basal amygdala nuclei to conditioned fear. *J Neurosci.* 31, 15481-15489.

Bayer, H.M., and Glimcher, P.W. (2005). Midbrain dopamine neurons encode a quantitative reward prediction error signal. *Neuron* 47, 129-141.

Cai, L.X., Pizano, K., Gundersen, G.W., Hayes, C.L., Fleming, W.T., Holt, S., Cox, J.M., Witten, I.B. (2020). Distinct signals in the medial and lateral VTA dopamine neurons modulate fear extinction at different times. *Elife* 9, e54936.

Chen APF, Chen L, Shi KW, Cheng E, Ge S, Xiong Q (2023). Nigrostriatal dopamine modulates the striatal-amygdala pathway in auditory fear conditioning. *Nat Commun.* 14, 7231.

Cohen, J.Y., Haesler, S., Vong, L., Lowell, B.B. and Uchida, N. (2012). Neuron-type-specific signals for reward and punishment in the ventral tegmental area. *Nature* 482, 85–88.

de Jong, J.W., Afjei, S.A., Pollak Dorocic, .I, Peck, J.R., Liu, C., Kim, C.K., Tian, L., Deisseroth, K., and Lammel, S. (2019). A neural circuit mechanism for encoding aversive stimuli in the mesolimbic dopamine system. *Neuron* 101, 133–151.

Eshel, N., Bukwich, M., Rao, V., Hemmelder, V., Tian, J., and Uchida, N. (2015). Arithmetic and local circuitry underlying dopamine prediction errors. *Nature* 525, 243-246.

Eshel, N., Tian, J., Bukwich, M., and Uchida, N. (2016). Dopamine neurons share common response function for reward prediction error. *Nat. Neuro.* 19, 479-486.

Groessl, F., Munsch, T., Meis, S., Griessner, J., Kaczanowska, J., Pliota, P., Kargl, D., Badurek, S., Kraitsy, K., Rassoulpour, A., Zuber, J., Lessmann, V., and Haubensak, W. (2018). Dorsal tegmental dopamine neurons gate associative learning of fear. *Nat. Neuro.* 21, 952–962.

Herry C, Ciocchi S, Senn V, Demmou L, Müller C, Lüthi A (2008). Switching on and off fear by distinct neuronal circuits. *Nature.* 454, 600-606.

Hunnicut B, Jongbloets BC, Birdsong WT, Gertz KJ, Zhong H, Mao T (2016) A comprehensive excitatory input map of the striatum reveals novel functional organization. *Elife* 5: e19103.

Kintscher M, Kochubey O, Schneggenburger R (2023). A striatal circuit balances learned fear in presence and absence of sensory cues. *Elife* 12, e75703.

Lammel, S., Ion, D.I., Roeper, J., Malenka, R.C. (2011). Projection-specific modulation of dopamine neuron synapses by aversive and rewarding stimuli. *Neuron.* 70, 855-862.

Matsumoto, M., Hikosaka, O. (2009). Two types of dopamine neurons distinctly convey positive and negative motivational signals. *Nature* 459, 837-841.

Menegas, W., Babayan, B.M., Uchida, N., and Watabe-Uchida, M. (2017). Opposite initialization to novel cues in dopamine signaling in ventral and posterior striatum in mice. *eLife.* 6, e21886.

Menegas, W., Akiti, K., Amo, R., Uchida, N., and Watabe-Uchida, M. (2018). Dopamine neurons projecting to the posterior striatum reinforce avoidance of threatening stimuli. *Nat. Neuro.* 21, 1421–1430.

Maren, S., Phan, K.L., Liberzon, I. (2013). The contextual brain: implications for fear conditioning, extinction and psychopathology. *Nat Rev Neurosci.* 14, 417-428. doi: 10.1038/nrn3492.

Nardoci MB, Lakunina AA, Henderling DC, Pedregon JC, Mohn JL, Jaramillo S (2022). Sound-evoked responses of distinct neuron classes from the tail of the striatum. *eNeuro.* 9, ENEURO.0201-22.2022.

Popescu, A.T., Saghyan, A.A., Paré, D. (2007). NMDA-dependent facilitation of corticostriatal plasticity by the amygdala. *Proc Natl Acad Sci U S A.* 104, 341-346.

Salinas-Hernández, X.I., Vogel, P., Betz, S., Kalisch, R., Sigurdsson, T., Duvarci, S. (2018). Dopamine neurons drive fear extinction learning by signaling the omission of expected aversive outcomes. *Elife* 7, e38818.

Salinas-Hernández, X.I., Zafiri, D., Sigurdsson, T., and Duvarci, S. (2023). Functional architecture of dopamine neurons driving fear extinction learning. *Neuron* 111, 3854–3870.

Schultz, W., Dayan, P., and Montague, P.R. (1997). A neural substrate of prediction and reward. *Science* 275, 1593-1599.

Sierra-Mercado D, Padilla-Coreano N, Quirk GJ (2011). Dissociable roles of prelimbic and infralimbic cortices, ventral hippocampus, and basolateral amygdala in the expression and extinction of conditioned fear. *Neuropsychopharmacology*. 36, 529-538.

Steinberg, E.E., Keiflin, R., Boivin, J.R., Witten, I.B., Deisseroth, K., and Janak, P.H. (2013). A causal link between prediction errors, dopamine neurons and learning. *Nat. Neurosci.* 16, 966-973.

Steinberg EE, Gore F, Heifets BD, Taylor MD, Norville ZC, Beier KT, Földy C, Lerner TN, Luo L, Deisseroth K, Malenka RC (2020). Amygdala-midbrain connections modulate appetitive and aversive learning. *Neuron*. 106, 1026-1043.e9.

Yuan, L., Dou, Y.N., and Sun, Y.G. (2019). Topography of Reward and Aversion Encoding in the Mesolimbic Dopaminergic System. *J Neurosci.* 39, 6472–6481.

Wilensky AE, Schafe GE, LeDoux JE (1999). Functional inactivation of the amygdala before but not after auditory fear conditioning prevents memory formation. *J Neurosci.* 19, RC48.

Reviewer comments:

Reviewer #1 (Remarks to the Author):

The authors have performed additional experiments and addressed my previous concerns. This is an important study that shows a critical role of dopamine in the tail of the striatum in a classic fear conditioning paradigm.

Author's Response: We are grateful for the Reviewer's positive evaluation of our manuscript.

Reviewer #2 (Remarks to the Author):

The authors have addressed all my concerns, and I am pleased to see they have conducted extensive new experiments to do so, including loss of function optogenetic assays and also anxiety tasks.

Author's Response: We are grateful for the Reviewer's positive evaluation of our manuscript.

Reviewer #3 (Remarks to the Author):

This revised manuscript by Zafiri is comprehensive in its assessment of dopamine prediction error encoding within SNc projections to tail of the striatum for associative fear learning. The study incorporates numerous cutting-edge techniques and addresses a fundamental question relating to dopamine in this context. The authors have adequately addressed prior reviewer concerns and provided numerous additional experiments and analyses that have significantly strengthened their study. I have no additional comments or concerns.

Author's Response: We are grateful for the Reviewer's positive evaluation of our manuscript.

Reviewer #4 (Remarks to the Author):

The authors did a great job to address all previous concerns. This is a great set of data to advance our knowledge of the tail of the striatum.

Author's Response: We are grateful for the Reviewer's positive evaluation of our manuscript.